# Perovskite-polymer composite cross-linker approach for highly-stable and efficient perovskite solar cells

Tae-Hee Han[1], Jin-Wook Lee[1], Chungseok Choi[1], Shaun Tan[1], Changsoo Lee[2], Yepin Zhao[1], Zhenghong Dai[1], Nicholas De Marco[1], Sung-Joon Lee [1], Sang-Hoon Bae[1], Yonghai Yuan[3], Hyuck Mo Lee[2], Yu Huang[1] & Yang Yang[1]

Manipulation of grain boundaries in polycrystalline perovskite is an essential consideration for both the optoelectronic properties and environmental stability of solar cells as the solution-processing of perovskite films inevitably introduces many defects at grain boundaries. Though small molecule-based additives have proven to be effective defect passivating agents, their high volatility and diffusivity cannot render perovskite films robust enough against harsh environments. Here we suggest design rules for effective molecules by considering their molecular structure. From these, we introduce a strategy to form macromolecular intermediate phases using long chain polymers, which leads to the formation of a polymer-perovskite composite cross-linker. The cross-linker functions to bridge the perovskite grains, minimizing grain-to-grain electrical decoupling and yielding excellent environmental stability against moisture, light, and heat, which has not been attainable with small molecule defect passivating agents. Consequently, all photovoltaic parameters are significantly enhanced in the solar cells and the devices also show excellent stability.

[1] Department of Materials Science and Engineering, and California NanoSystems Institute, University of California, Los Angeles, CA 90095, USA. [2] Department of Materials Science and Engineering, Korea Advanced Institute of Science and Technology, 291 Daehak-ro, Yuseong-gu, Daejeon 34141, Republic of Korea. [3] Solargiga Energy Holdings Limited, Hong Kong 999077, China. These authors contributed equally: Tae-Hee Han, Jin-Wook Lee. Correspondence and requests for materials should be addressed to Y.Y. (email: yangy@ucla.edu)

Metal halide perovskites have been used in various optoelectronic applications, such as photodetectors[1], light-emitting diodes[2,3], solar cells[4–9], X-ray imaging[10], and lasing[11] due to their high absorption coefficients[12], long-range charge carrier diffusion lengths[8,13], and high photoluminescence (PL) quantum yield[2]. Since the first efficient solid-state perovskite solar cell was reported in 2012[5], a lot of attempts to understand the photophysical properties of perovskites and to improve the photovoltaic performance of perovskite-based solar cells have been made. Recent improvements achieved via compositional[14,15], morphological[16,17], and interfacial engineering[18] have resulted in a rapid increase in the power conversion efficiency (PCE) of metal halide perovskite solar cells, making them a strong candidate to compete against the more well-developed inorganic semiconductors based on high-vacuum processes, such as silicon, gallium arsenide or copper-indium-gallium-selenide[19].

Manipulation of defective grain boundaries in polycrystalline perovskite films is crucial to maximize both the optoelectronic properties and stability of the film and the corresponding devices[20,21]. The superior optoelectronic properties of single crystal perovskites over widely adopted polycrystalline perovskite thin films imply that grain boundaries play a critical roles on the optoelectronic properties of the film. For example, carrier diffusion lengths of single crystal perovskites and polycrystalline thin films are more than 100 $\mu$m and less than 10 $\mu$m, respectively, whereas trap densities for single crystal perovskites are between $10^9$ and $10^{10}$ cm$^{-3}$ as compared to $10^{16}$ to $10^{18}$ cm$^{-3}$ for polycrystalline thin film[8,13,22,23]. The characteristic grain boundaries of the polycrystalline thin films were found to function as trap states and further act as vulnerable spots to trigger the degradation of the materials and its physical properties[21,22,24]. Because metal halide perovskite films are deposited via solution processes and crystallizes at low temperatures, a lot of structural defects exist along the grain boundaries of polycrystalline perovskite films. Grain boundaries that have dangling bonds can provide migration paths for ions and can become charge carrier trap centers and cause non-radiative recombination, which can significantly degrade charge carrier transport and the photophysical properties of the perovskite film[25–27]. Furthermore, defective grain boundaries are more vulnerable to heat and moisture degradation which propagates inwards into the grain interiors from the boundaries to induce the physical and electrical decoupling of individual grains, thus reducing device performance. Therefore, it is important for polycrystalline perovskite films to meet several requirements for optoelectronic applications: (1) high crystallinity and large-sized crystal growth to minimize grain boundaries and structural defects at both grain interiors and boundaries, (2) effective defect passivation at grain boundaries, and (3) cross-linking of individual crystal grains for high stability against harsh environmental stresses.

Precursors for metal halide perovskites such as Pb(II) halides (e.g., PbI$_2$, PbBr$_2$ or PbCl$_2$) or organic halides (e.g., CH$_3$NH$_3$I, HC(NH$_2$)$_2$I) are known to be Lewis acids[28,29]. Reaction of a Lewis acid with a Lewis base leads to either a redox reaction or an adduct formation, the latter of which is composed of the acid and base linked by a dative bond (i.e., shared electrons that originate from the Lewis base)[20,24]. The intermediate adduct phase formed by such Lewis base–acid reaction facilitates the homogeneous crystal growth of the perovskite due to the additional Lewis base removal process from the adduct film, which retards the formation rate constant for the perovskite[24]. This intermediate phase method has been used in perovskite solar cells, but the use of Lewis bases have been restricted to polar aprotic small molecules, such as dimethyl sulfoxide (DMSO), urea, and N-methyl-2-

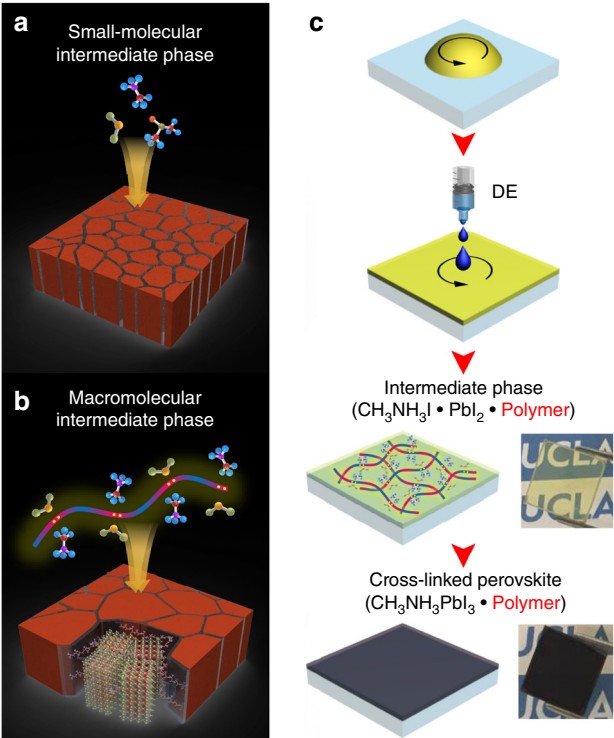

**Fig. 1** Schematic illustrations of the perovskite film formation and fabrication steps. Perovskite grain growth induced by **a** conventional small molecular and **b** macromolecular intermediate phase. **c** Fabrication steps for the macromolecular intermediate phase-mediated crystal growth of the perovskite film (optical images: transparent adduct film and perovskite film on glass)

pyrrolidone (NMP)[20,24]. The small molecule Lewis bases form small molecular adducts with individual molecules of the perovskite precursors (Fig. 1a). Also, small molecule-based defect passivating agents that have lone pair electrons on oxygen, sulfur, or nitrogen (e.g., pyridine[30], thiophene[30], and fullerenes[31]) have been used to improve the photophysical properties of perovskites by coordinating with defect sites at grain boundaries. However, the high degree of volatility and high diffusion coefficients of small molecules likely pose difficulties in incorporating them into practical devices operated under harsh environments, such as high temperature, humidity, electric field, and strong light. For these reasons above, there have been several attempts to use polymeric additives[21,31–37] or cross-linking small molecular agents[38,39] incorporated into the perovskite active layer as a crystal growth template, defect passivating molecule or crystal cross-linker. Long-chain polymers could be immobilized after crystallization of the perovskite, while small molecule additives have substantial diffusion and drift mobility in the perovskite film during operation.

Here, we introduce an inter-grain cross-linking strategy, induced by a macromolecular intermediate phase with a polymeric Lewis base, for simple fabrication of highly efficient and stable perovskite solar cells. Repeating units of the long-chained polymeric Lewis base can possibly form Lewis acid–base adducts with a series of perovskite precursor molecules, so it forms a macromolecular intermediate phase with a much larger degree of coordination and long-range molecular ordering along the polymer chain (Fig. 1b). This subsequently leads to the formation of enlarged, cross-linked perovskite grains, which are also defect-passivated by the remnant polymeric Lewis bases.

## Results

**Macromolecular Lewis acid–base intermediate phase.** Ethylene carbonate (EC [$C_3H_4O_3$], 5.41 $D$) and propylene carbonate (PC [$C_4H_6O_3$], 5.57 $D$) as small molecule Lewis bases with different permanent dipoles and poly(propylene carbonate) (PPC, [$C_4H_6O_3$]$_n$) as a polymeric Lewis base were investigated in order to establish how the molecular dipole and molecular structure of Lewis bases affect the formation of the intermediate phase and growth kinetics of perovskite. PPC is a linear copolymer of carbon dioxide and propylene oxide. PC, the repeating unit of PPC, possesses a high molecular dipole moment (5.57 $D$), which enables it to be dissolved in polar solvents (e.g., N,N-Dimethylformamide (DMF) or DMSO) and to have strong molecular interactions with the Lewis acidic perovskite precursors of $CH_3NH_3PbI_3$, i.e., the methylammonium cations ($CH_3NH_3^+$) and iodoplumbate anions. Carbonate groups (C(=O)(O–)$_2$) in the PPC can be considered as a Lewis base that donates the lone pair electrons on its oxygen to form a strong Lewis acid–base adduct with the perovskite precursors. The high molecular dipole in PC enhances its interactions with $CH_3NH_3^+$ that has a relatively high dipole moment (2.3 $D$). The long chain polymer does not evaporate unlike DMSO which has a comparatively high vapor pressure[24]. Thus, PPC remains in the perovskite film even after crystallization at high temperatures (Supplementary Figure 1). Remnant polymeric Lewis bases can donate lone pair electrons from the oxygen atom to coordinate with perovskite crystal defects such as $Pb^{2+}$ or $NH_3^+$ at the grain boundaries[20,21]. On the other hand, PPC is insoluble in water and its repeating unit (PC) has a hydrophobic nature[40]. Therefore, PPC functions both as a cross-linker and passivation agent to enhance the moisture resistance of perovskite polycrystals in ambient conditions.

A film formation process is schematically depicted in Fig. 1c. DMF, which has a weak Lewis basicity, was used as a base solvent to dissolve the perovskite precursors ($CH_3NH_3I$ and $PbI_2$) and their adduct complexes. Equimolar DMSO, the most widely used Lewis basic polar aprotic solvent, was also used to form a 1:1 adduct with the perovskite precursors. A small amount of polymeric Lewis base was included into the precursor solution, and formed an intermediate phase having long-range molecular ordering with the precursors (i.e., $CH_3NH_3I \cdot PbI_2 \cdot DMF \cdot DMSO \cdot$ Polymeric Lewis base). Diethyl ether (DE) was dropped during spin-casting of the perovskite precursor solution, which washes DMF away from the film. As a result, a $CH_3NH_3I \cdot PbI_2 \cdot DMSO \cdot$ Polymeric Lewis base adduct remains in the film. DMSO also evaporates during high temperature annealing, resulting in perovskite crystals modified by polymers (i.e., $CH_3NH_3I \cdot PbI_2 \cdot DMSO \cdot$ Polymeric Lewis base adduct+heat→$CH_3NH_3PbI_3 \cdot$ Polymeric Lewis base) (Supplementary Figure 1).

Fourier transform infrared (FTIR) spectroscopy was conducted on the synthesized adduct powders to investigate the interactions between the Lewis bases and the perovskite precursors (Fig. 2, and Supplementary Figure 2). The N–H stretch ($\nu_1$(asym) = 3161 cm$^{-1}$, $\nu_2$(sym) = 3124 cm$^{-1}$) of the adduct powder without a Lewis base were shifted when DMSO was included, which indicates a chemical interaction of the cationic charge $NH_3^+$ in $CH_3NH_3I$ with the Lewis base[41,42] (Fig. 2a). When the Lewis bases with higher dipole moments than DMSO were included, a further shift to ($\nu_1$(asym) = 3199 cm$^{-1}$, $\nu_2$(sym) = 3162 cm$^{-1}$) was observed. Compared to the FTIR spectra for pure Lewis bases, all the C=O stretch vibration peaks of the Lewis bases were redshifted in the synthesized $CH_3NH_3I \cdot PbI_2 \cdot DMSO \cdot$ Lewis base adduct powder (Fig. 2b–d). The C=O stretch vibration peak for pure PPC was around 1760 cm$^{-1}$ [43], and it was redshifted to 1743 cm$^{-1}$. Because, the vibrational frequency is proportional to the square root of the force constant in the harmonic motion of a diatomic model, a decrease in the C=O stretch vibration of the Lewis bases means that the bond strengths of carbon and oxygen decreased upon formation of the $CH_3NH_3I \cdot PbI_2 \cdot DMSO \cdot$ Lewis base adduct, suggesting strong chemical interaction between $NH_3^+$ and the oxygen in the Lewis base. Density functional theory (DFT) calculations were also undertaken to investigate the interaction energies and the most stable molecular configurations for the different Lewis bases additions. It was observed from the calculations that both the oxygen (S=O) in DMSO and the oxygen (C=O) in the carbonate group of the Lewis bases interact with the ammonium cation $CH_3NH_3^+$. Furthermore, higher interaction energies were calculated for the cases where Lewis bases with higher dipole moments were added as compared to the case where DMSO was added (interaction energy: $-1.345$ eV for $CH_3NH_3^+$-DMSO, $-2.225$ eV for $CH_3NH_3^+$-DMSO-EC, and $-2.256$ eV for $CH_3NH_3^+$-DMSO-PC) (Supplementary Figure 3 and Supplementary Table 1). The interaction energies between $CH_3NH_3^+$ and PPC with different numbers of repeating units were also calculated and was shown to be higher for longer PPC chain lengths (e.g., the interaction energy of $CH_3NH_3^+$-DMSO-PPC with four repeating units was $-2.283$ eV) (Supplementary Figure 4 and Supplementary Table 2). Thermogravimetric analysis (TGA) on the PPC·perovskite precursor adducts confirmed this as seen by the substantially increased sublimation temperatures due to the formation of the polymeric adduct (Supplementary Figure 1b and Supplementary Note 1). According to both experimental results and theoretical calculations, the donation of the lone pair electrons on the oxygen of the Lewis bases to $CH_3NH_3^+$ and $PbI_2$ results in the formation of a $CH_3NH_3I \cdot PbI_2 \cdot DMSO \cdot$ Lewis base adduct, and the addition of a Lewis base with a higher permanent dipole can more chemically stabilize the adduct formation. At a constant temperature, the crystallization rate parameter is exponentially proportional to the inverse of the activation energy ($E_a$)[44]. An enhanced Lewis acid–base interaction due to the addition of the Lewis bases increases $E_a$ for crystallization, which decreases the number of nucleations and slows down crystal growth[20,24,45]. Therefore, such enhanced Lewis acid–base interaction between the perovskite precursors and the Lewis base realizes the controlled crystallization of the perovskite.

To determine how the polymeric Lewis bases influence perovskite crystal growth, X-ray diffraction (XRD) studies were carried out on the synthesized adduct powders and perovskite films (Fig. 2e–h). The XRD spectra of the synthesized adduct powders with DMSO or PPC exhibited peaks lower than 10° due to the longer interplanar distances of the adducts relative to that of the pure perovskite[14]. It was observed that most of their peaks disappeared after crystallization. Thus, the XRD peaks of the adduct powder provide information on the molecular ordering of the intermediate phases (Fig. 2e, f). Polymeric Lewis bases dramatically increased the XRD intensities of the intermediate phase, as compared to the case with DMSO. Polymeric Lewis bases reduced the full width at half maximum (FWHM) of the main peak at 8° from 0.134 to 0.115, indicating an enhanced molecular ordering of the intermediate phase induced by the long chain polymer. Additionally, a significant increase in the XRD peak intensities at high angles was also seen, further confirming the improved long-range molecular ordering of the intermediate phase (Fig. 2e, f). After crystallization, peak intensities of perovskite films also increased with polymeric Lewis bases. The addition of PPC increased the (110) peak intensity by approximately 30% and reduced its FWHM from 0.142 to 0.121 (Fig. 3g, h) while the peak positions remained unchanged. Atomic force microscopy (AFM) was used to confirm the enlarged grain sizes induced by the polymer (Fig. 2i–l). Even though small molecular Lewis bases (EC and PC) likewise increased the perovskite

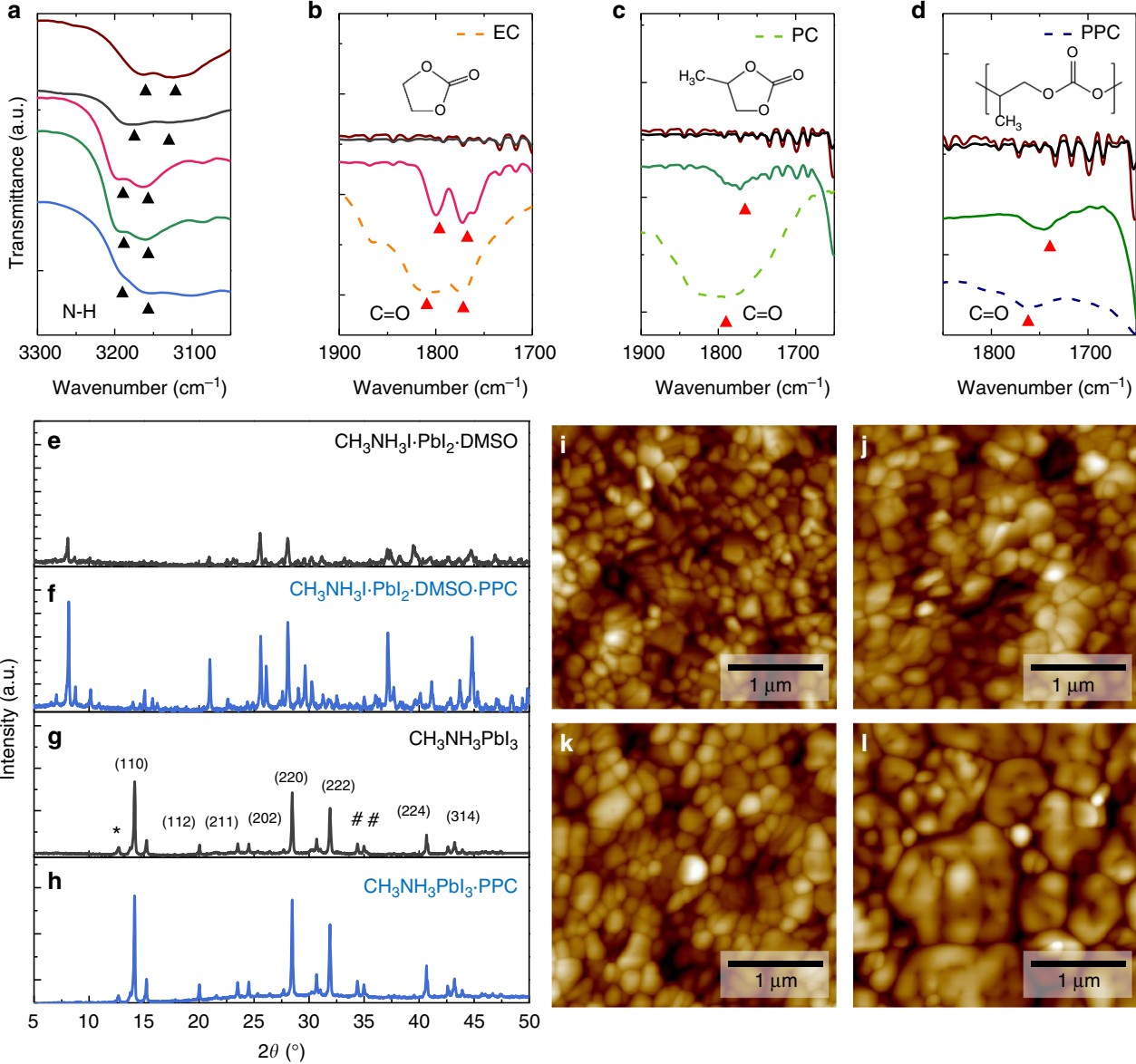

**Fig. 2** Formation of the macromolecular intermediate phase. Fourier transform infrared (FTIR) spectra of synthesized adduct powders of CH$_3$NH$_3$I and PbI$_2$ without (brown) and with the addition of Lewis bases (DMSO (black), EC (red), PC (green), and PPC (blue)). The FTIR spectra for the **a** N-H stretch (black arrows) and **b–d** C=O stretch (red arrows) (brown: without Lewis base, black: CH$_3$NH$_3$I·PbI$_2$·DMSO, red: CH$_3$NH$_3$I·PbI$_2$·DMSO·EC, green: CH$_3$NH$_3$I·PbI$_2$·DMSO·PC, and blue solid line: CH$_3$NH$_3$I·PbI$_2$·DMSO·PPC) (inset: chemical structure of **b** EC, **c** PC, and **d** PPC). X-ray diffraction spectra of synthesized **e** CH$_3$NH$_3$I·PbI$_2$·DMSO and **f** CH$_3$NH$_3$I·PbI$_2$·DMSO·PPC adduct powder. X-ray diffraction spectra of **g** CH$_3$NH$_3$PbI$_3$ and **h** CH$_3$NH$_3$PbI$_3$·PPC perovskite films, Atomic force microscope topographic images of CH$_3$NH$_3$PbI$_3$ **i** without additive, **j** with PC, **k** with PPC (0.1 wt%), and **l** with PPC (0.3 wt%)

crystallinity and grain size with increasing molecular dipole moment (Fig. 2i, j and Supplementary Figures 5, 6), crystal grains grown with PPC were much larger than those grown with the small molecules (Fig. 2i–l).

**Inter-grain cross-linking of the perovskite**. Beyond large-sized crystal grain growth, the macromolecular intermediate phase method additionally enables the inter-grain cross-linking of the perovskite (Fig. 3). Scanning electron microscopy (SEM) images clearly show much larger perovskite grains with the addition of PPC than those without the Lewis base (Fig. 3a, b), and most of the enlarged perovskite grains were merged with one another via cross-linking (highlighted region in Fig. 3b). Transmission electron microscopy (TEM) analysis was used to confirm the cross-

linking between perovskite grains. The TEM image in Fig. 3c visualizes the cross-linking of two perovskite grains which were scratched out from the perovskite film, and dispersed in an anti-solvent (chloroform). Figure 3d is a magnified TEM image from the highlighted region of Fig. 3c (bridge between two grains). We analyzed the two highlighted regions of Fig. 3d: (1) the interior and (2) the edge of the cross-linker. Fast Fourier transform (FFT) analysis in the interior of the bridge (highlighted region 1) shows an interplanar spacing of 3.1 Å, which matches well with the (110) reflection of cubic CH$_3$NH$_3$PbI$_3$ (Fig. 3e). On the other hand, the inverse FFT image magnified on highlighted region 2 clearly shows the formation of a polymer-perovskite composite bridge between two grains (Fig. 3f). Conventional passivating molecules likely form secondary pure organic phases between the perovskite grains[20,36], and these organic phases do not absorb

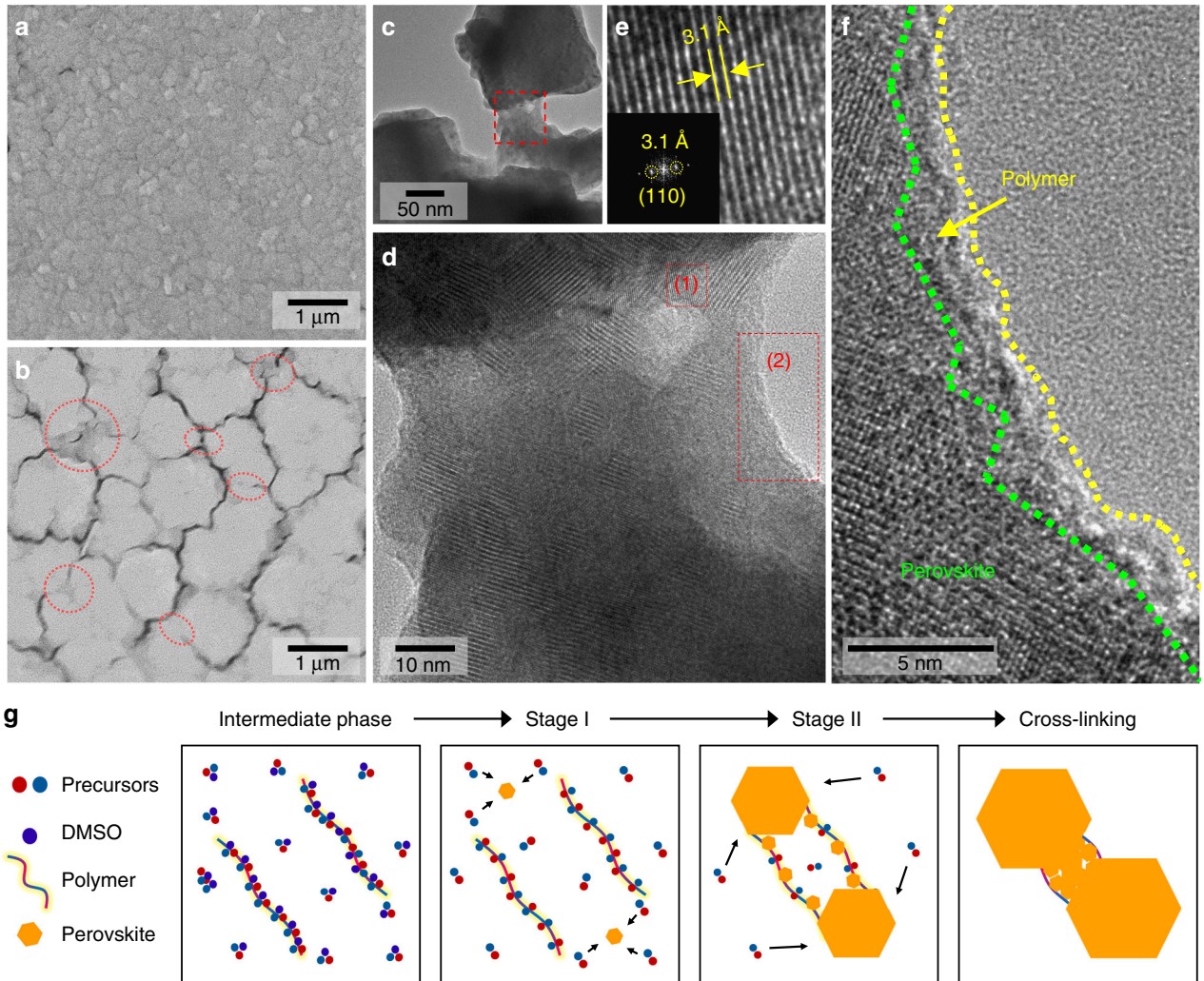

**Fig. 3** Microstructure of the cross-linked perovskite-polymer film. Scanning electron microscopy images of the **a** CH$_3$NH$_3$PbI$_3$ and **b** CH$_3$NH$_3$PbI$_3$·PPC film. Transmission electron microscopy (TEM) images of the **c** cross-linked perovskite grains at lower magnification and **d** perovskite-polymer composite cross-linker between adjacent grains (highlighted region in Fig. 3c). Inverse Fast Fourier transform (IFFT) images of the region **e** (1): within the perovskite-polymer bridge (Inset: FFT analysis of (1)) and **f** (2): boundary of the perovskite-polymer composite cross-linker. **g** Schematic drawing of macromolecular intermediate phase induced crystallization and inter-grain cross-linking

light and can possibly electrically decouple the perovskite crystals. In contrast, the polymer-perovskite composite interconnector reported here could minimize the electrical decoupling or insulation between the perovskite grains due to its unique composition.

The following mechanism is suggested to rationalize the inter-grain cross-linking (Fig. 3g). DMSO evaporates at a low temperature (70 °C), and supersaturates the perovskite precursor solution (Supplementary Figures 1, 7 and 8). Conventional small molecule intermediate phases without polymers have high nucleation rates, and thus form a large number of nuclei, such that the nuclei impinge on one another and impede further grain growth, resulting in small grains (Supplementary Figure 8a). Nucleation rate is exponentially proportional to the inverse of the energy barrier to form a stable nucleus ($\Delta G^*$) and diffusion ($\Delta E_D$). Since a tightly bound polymeric adduct increases both $\Delta G^*$ and $\Delta E_D$, the macromolecular intermediate phase decreases the nucleation probability and thus slows down the rate of nucleation, thereby decreasing the number of nuclei (Supplementary Figures 1, 4 and Supplementary Table 2). In other words, a relatively small amount of nuclei nucleate without high energy barriers and subsequently grow continuously without impingement to form large grains. At an elevated temperature (i.e., stage II), the bounded perovskite precursors within the polymeric adducts can be crystallized by overcoming the increased activation energy barrier for nucleation, and such crystals grown in the proximity of long-chain polymers are immobilized by the surrounding polymer chains and interconnect large grains with polymers (Fig. 3g). The ability to form an appropriate interaction between the polymer and the perovskite precursors crucially affects the perovskite crystal growth and the resulting electrical properties of the perovskite film. The Lewis basicity and molecular structure of a polymer's repeating functional group also affects the formation of the macromolecular intermediate phase with the perovskite precursor. We compared the effects of different functional groups of polymer on the perovskite crystal growth and subsequent electrical properties (Supplementary Figures 9–14 and Supplementary Notes 2–4) and observed that the polymeric Lewis base increased the electrical conductivity of the perovskite film while in contrast, the polymeric acid severely interrupted the spatial electrical conductance of the perovskite grains, which indeed resulted in a huge drop in the photovoltaic performance of the devices with a high dose of the polymeric molecules.

**Defect passivation effect**. The presence of the Lewis bases in the perovskite films was confirmed from the reflection FTIR spectra of the films (Fig. 4a, b). Characteristic C=O stretch peaks around 1789 cm⁻¹ were observed for the perovskite films with the Lewis bases (black arrows). The remnant Lewis base molecules in the films interacted with the perovskites through a coordinative covalent bond to passivate defects at the grain boundaries[20]. To elucidate the effects of these interactions on the optoelectronic properties of the films, the binding energies of each Lewis base bonded to the perovskite crystal were estimated using DFT calculations (Fig. 4c). The binding energies were calculated to be −19.4 kJ mole⁻¹, −31.5 kJ mole⁻¹ and −34.5 kJ mole⁻¹ for EC, PC, and PPC, respectively. Considering that the grain boundary defects are predominantly iodide (I⁻) vacancies[46], it is speculated that the interaction between the Lewis bases and the perovskite could be a dipole–ion interaction between the lone pair electrons of the Lewis bases and under-coordinated Pb atoms at the grain boundaries. Therefore, the stronger binding energy when PC was added in comparison to the case with EC can be correlated with the stronger dipole moment of the former (5.57 $D$) relative to the latter (5.42 $D$). Although the repeating molecule of PPC has a lower dipole moment, the partial atomic charge on the oxygen atom in the molecule was calculated to be higher than those of EC and PC, such that the extent of charge transfer was also higher than those of the small molecules (charge transfer from Lewis base molecule to surface: 0.008e for EC, 0.014e for PC, and 0.030e for PPC) (Supplementary Figure 15 and Supplementary Table 3).

Charge carrier behavior in the perovskite films was investigated by PL spectroscopy (Fig. 4d, e). The steady-state PL spectra did not show any noticeable changes in peak positions, indicating that the Lewis bases did not affect the optical bandgap of the perovskite. Notably, the peak PL intensity was enhanced from $4.07 \times 10^5$ (bare $CH_3NH_3PbI_3$) to $6.96 \times 10^5$, $9.67 \times 10^5$ and $12.10 \times 10^5$ with the use of EC, PC, and PPC, respectively (Fig. 4d, and Supplementary Figure 16). Time-resolved PL decay profiles were measured in which empty circles indicate measured data while solid lines indicate fitted curves (Fig. 4e). The curves are fitted using a bi-exponential decay model ($y = A_1\exp[-(x-x_0)/\tau_1] + A_2(\exp[-(x-x_0)/\tau_2])$), and the fitted parameters are summarized in Table 1. The inset of Fig. 4e shows average PL lifetimes of the films. The relatively faster decay components ($\tau_1$ less than

**Table 1 Fitted parameters for the time-resolved photoluminescence (PL) decay profiles**

|  | $CH_3NH_3PbI_3$ | w/EC | w/PC | w/PPC |
|---|---|---|---|---|
| $A_1$ | 1018.8 | 431.7 | 447.7 | 510.9 |
| Proportion (%) | 74.8 | 43.0 | 44.1 | 47.2 |
| $\tau_1$ (ns) | 7.9 | 15.0 | 17.8 | 14.3 |
| $A_2$ | 343.9 | 572.8 | 568.4 | 571.6 |
| Proportion (%) | 25.2 | 57.0 | 55.9 | 52.8 |
| $\tau_2$ (ns) | 49.6 | 87.7 | 99.4 | 120.4 |
| $\tau_{avg}$ (ns) | 18.4 | 56.5 | 63.4 | 70.3 |

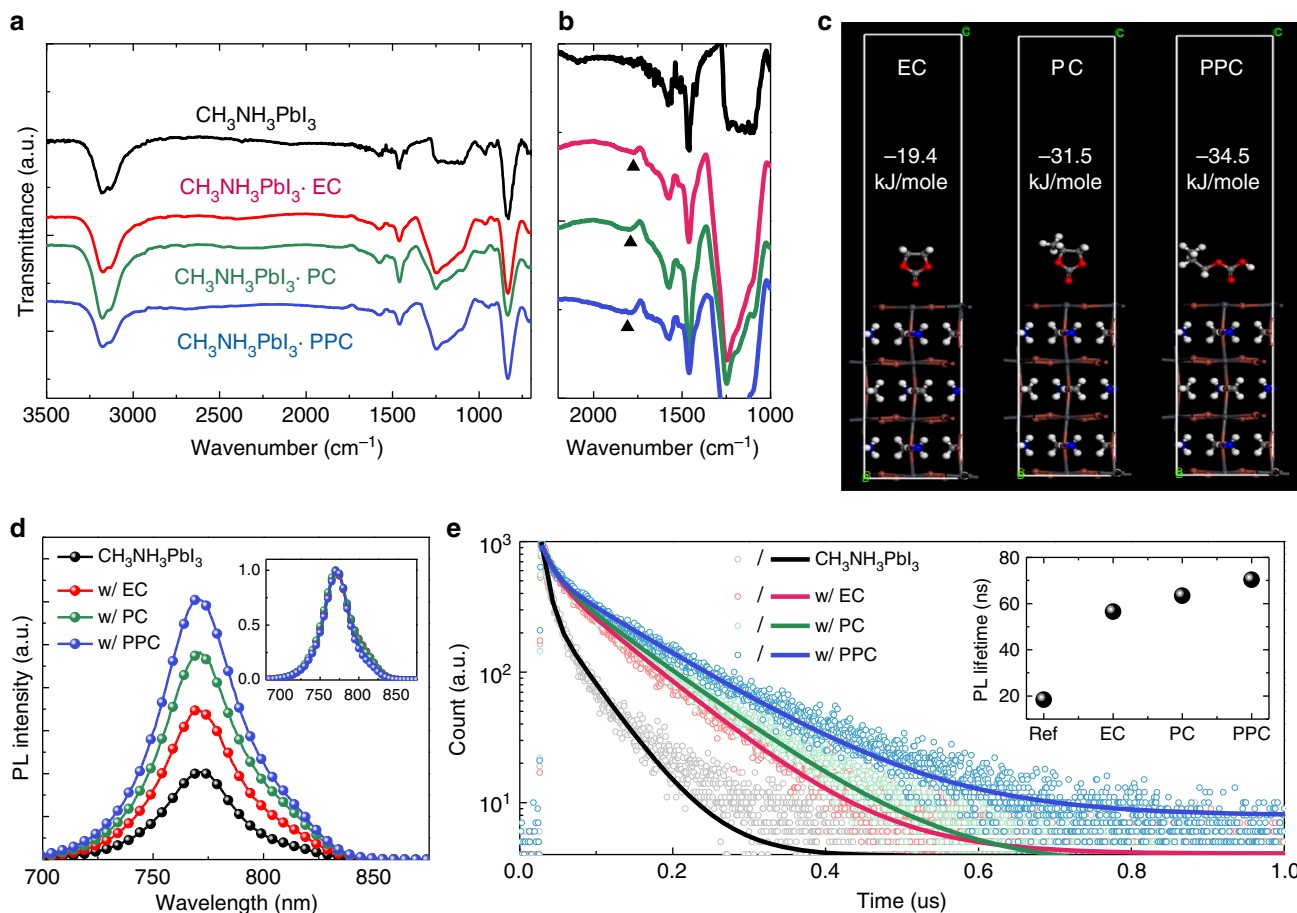

**Fig. 4** Defect passivation by the polymeric Lewis base. **a** Reflective Fourier transform infrared (FTIR) spectra of the perovskite films without and with various types of Lewis bases (EC, PC, and PPC), **b** The FTIR spectra of the C=O stretch (black arrows), **c** Most favorable Pb²⁺ perovskite surface adsorption configurations of the Lewis bases and their calculated binding energies, **d** Photoluminescence (PL) (Inset: normalized PL spectra) and **e** time-resolved PL spectra of perovskite films without and with Lewis bases (Inset: PL lifetimes fitted from the time-resolved PL spectra)

20 ns) are attributed to charge carrier trapping defect states, while the slower decay components ($\tau_2$ more than 20 ns) are assigned to bimolecular radiative recombination in the bulk crystals[22]. With addition of EC and PC, the proportion of the fast decay component ($\tau_1$) is significantly reduced from 74.8% to 43.0% (EC) and 44.1% (PC) with a slower trapping time constant (15.0 ns for EC and 17.8 ns for PC) compared to that of the bare perovskite film (7.9 ns), which is indicative of a reduced defect density in the perovskite. Furthermore, $\tau_2$ was enhanced from 49.6 ns (bare $CH_3NH_3PbI_3$) to 87.7 ns (EC) and 99.4 ns (PC), suggesting that the Lewis bases also elongate charge carrier lifetimes within the bulk crystal. As a result, the average PL lifetime was greatly enhanced from 18.4 ns to 56.5 ns (EC) and 63.4 ns (PC). The lower defect density with reduced grain boundaries passivated by EC and PC might be the origin of the reduced charge carrier trapping while the elongated $\tau_2$ can be ascribed to larger crystallite sizes as observed by the XRD and AFM measurements (Fig. 2). Addition of PPC, instead of the small molecules, further elongated the PL lifetime. The longer PL lifetime with PPC as compared to the cases with the small molecular Lewis bases can be related to the larger crystallite sizes and higher binding energy of PPC. It is worth noting that the $\tau_2$ with PPC (120.4 ns) is considerably higher than those of the films with EC (87.7 ns) and PC (99.4 ns). The larger $\tau_2$ indicates a longer bulk carrier lifetime owing to the much larger-sized crystal grains and the inter-grain cross-linking induced by the long-range ordered intermediate phase as observed in the AFM and TEM analysis (Figs. 2, 3).

**Environmental stability and photovoltaic performances**. The environmental stability of the perovskite films modified using the polymeric Lewis bases was assessed against a control in different kinds of harsh environmental stresses that degrade perovskites (i.e., moisture, light, and heat) (Fig. 5a–c). A moisture stability test was conducted in an isolated chamber with $70 \pm 5\%$ relative humidity (RH) for 150 h. Separately, the films were kept under constant light illumination at AM 1.5G one-sun for 2 h to investigate the influence of light on the perovskite films. Also, thermal stability was tested by heating the films at 100 °C in a nitrogen atmosphere for 66 h. For each of the three environmental stress tests, the bare $CH_3NH_3PbI_3$ films were observed to severely decompose to $PbI_2$ in each case, confirmed by a substantial increase in the XRD peaks of the films at around 12.5° and also by a decreased absorption of the films (Fig. 5a–c, and Supplementary Figures 17–19). In contrast, the perovskite films with PPC retained relatively high amounts of $CH_3NH_3PbI_3$ compared to that of the bare perovskite films. With increasing amount of PPC added, the degradation of the $CH_3NH_3PbI_3$ films was further retarded, supporting the conclusion that the improved environmental stability originated from the addition of the PPC and their inter-grain cross-linking effect. The photovoltaic performances of the cross-linked $CH_3NH_3PbI_3 \cdot$PPC solar cells also exhibited a much superior resistance against the harsh environmental conditions (Supplementary Figures 20–22 and Supplementary Note 5), and thus demonstrates excellent practical environmental stability of the perovskite solar cells.

Photovoltaic performance of the perovskite solar cells with different Lewis base additions were compared as shown in Fig. 5d and Table 2. The devices incorporated a planar heterojunction structure with an architecture of ITO/SnO2/perovskite/spiro-MeOTAD/Ag. The averaged photovoltaic performances of reference cells (solar cells with bare $CH_3NH_3PbI_3$) was short-circuit current density ($J_{SC}$): $21.79 \pm 0.39$ mA cm$^{-2}$, open-circuit voltage ($V_{OC}$): $1.084 \pm 0.019$ V, fill factor (FF): $0.719 \pm 0.022$, and power conversion efficiency (PCE): $16.97 \pm 0.44\%$ (Fig. 5d and

Table 2). All the photovoltaic parameters were enhanced with incorporation of the Lewis bases (Fig. 5d and Supplementary Figures 23–25). The $J_{SC}$ was marginally enhanced from $21.79 \pm 0.39$ mA cm$^{-2}$ to $22.05 \pm 0.28$ mA cm$^{-2}$ (1.2% improvement) and $22.38 \pm 0.12$ mA cm$^{-2}$ (2.7% improvement), with the addition of EC and PC, respectively. A comparatively greater improvement in $V_{OC}$ and FF was observed where the $V_{OC}$ was increased from $1.084 \pm 0.019$ V to $1.115 \pm 0.019$ V (3.0% improvement with EC) and $1.118 \pm 0.011$ V (3.2% improvement with PC) while the FF was enhanced from $0.719 \pm 0.022$ to $0.759 \pm 0.013$ (5.6% improvement with EC) and $0.763 \pm 0.014$ (6.1% improvement with PC). As a result, the PCE was improved from $16.97 \pm 0.44\%$ to $18.65 \pm 0.51\%$ (9.8% improvement with EC) and $19.09 \pm 0.44\%$ (12.4% improvement with PC). The improved $V_{OC}$ and FF with EC and PC possibly originated from the reduced charge carrier recombination as confirmed by the PL studies. Also correlated with the superior PL lifetime, the devices with PPC showed the most pronounced improvement in $V_{OC}$ (to $1.129 \pm 0.008$ V, 6.7% improvement) and FF (to $0.767 \pm 0.013$, 4.2% improvement). As a result, the average PCE was improved by 14.0% from $16.97 \pm 0.44\%$ to $19.35 \pm 0.43\%$. As can be seen in Fig. 5e, a highest PCE of 20.06% ($J_{SC}$: 22.81 mA cm$^{-2}$, $V_{OC} = 1.131$ V, FF: 0.778) was achieved with the addition of PPC while the best device with bare $CH_3NH_3PbI_3$ reached 17.88% ($J_{SC}$: 22.43 mA cm$^2$, $V_{OC} = 1.083$ V, FF: 0.736). The improved $J_{SC}$ can be correlated with an improved absorbance at longer wavelengths, which is likely due to an enhanced light scattering by the larger grains[20] (Supplementary Figure 25). The stabilized PCEs were also measured to be 19.48% for the device with PPC and 16.94% with bare $CH_3NH_3PbI_3$ (Supplementary Figure 25).

To further confirm that the addition of the polymeric Lewis base had enhanced the ambient stability of the solar cell, the PCE evolution of the unencapsulated devices stored under ambient conditions without illumination was monitored. After 1008 h of storage, the average PCE of the bare $CH_3NH_3PbI_3$ devices were significantly degraded by 64.3%; whereas, the devices with PPC maintained their initial PCEs without any noticeable change (Fig. 5f). To demonstrate the universality of our cross-linking approach, we examined the operational behavior of the devices based on various kinds of 'A'-site cations. To evaluate their operational stability, all the devices were encapsulated under a nitrogen atmosphere, and exposed to continuous illumination ($90 \pm 10$ mW, without UV filter) under open-circuit condition (Supplementary Figures 26, 27, and Supplementary Table 4). PPC effectively elongated the operational lifetime of all the perovskite solar cells regardless of their 'A'-site cation composition. Most of the devices showed rapid initial decay followed by slower degradation with an almost linear profile, but the devices with PPC demonstrated a less severe initial decay. Given that the initial rapid decay can be related to the migration of ion and charged defects[47], the reduced initial decay regime can be attributed to a decreased charged defect density and an increased activation energy for ion migration as a result of the PPC-mediated crystallization. Severe ion migration in perovskite solar cells results in $J$–$V$ hysteresis, and along with the possible compound formation with the electrode, could accelerate the degradation of a device during operation[48–50]. Indeed, PPC effectively increased the activation energy for ion migration in the perovskite (Supplementary Figure 28 and Supplementary Note 6), resulting in reduced hysteresis than that of bare $CH_3NH_3PbI_3$ (Supplementary Figure 29). Furthermore, the addition of PPC slowed down the subsequent linear decay regime as well. From this, we extracted the $T_{80}$ (time with which the PCE decays to 80% of its initial value) from the post-burn-in regime (linear decay regime) for all the solar cells fabricated in this work and summarized this in Supplementary Table S4. All the calculated $T_{80}$ lifetimes of the

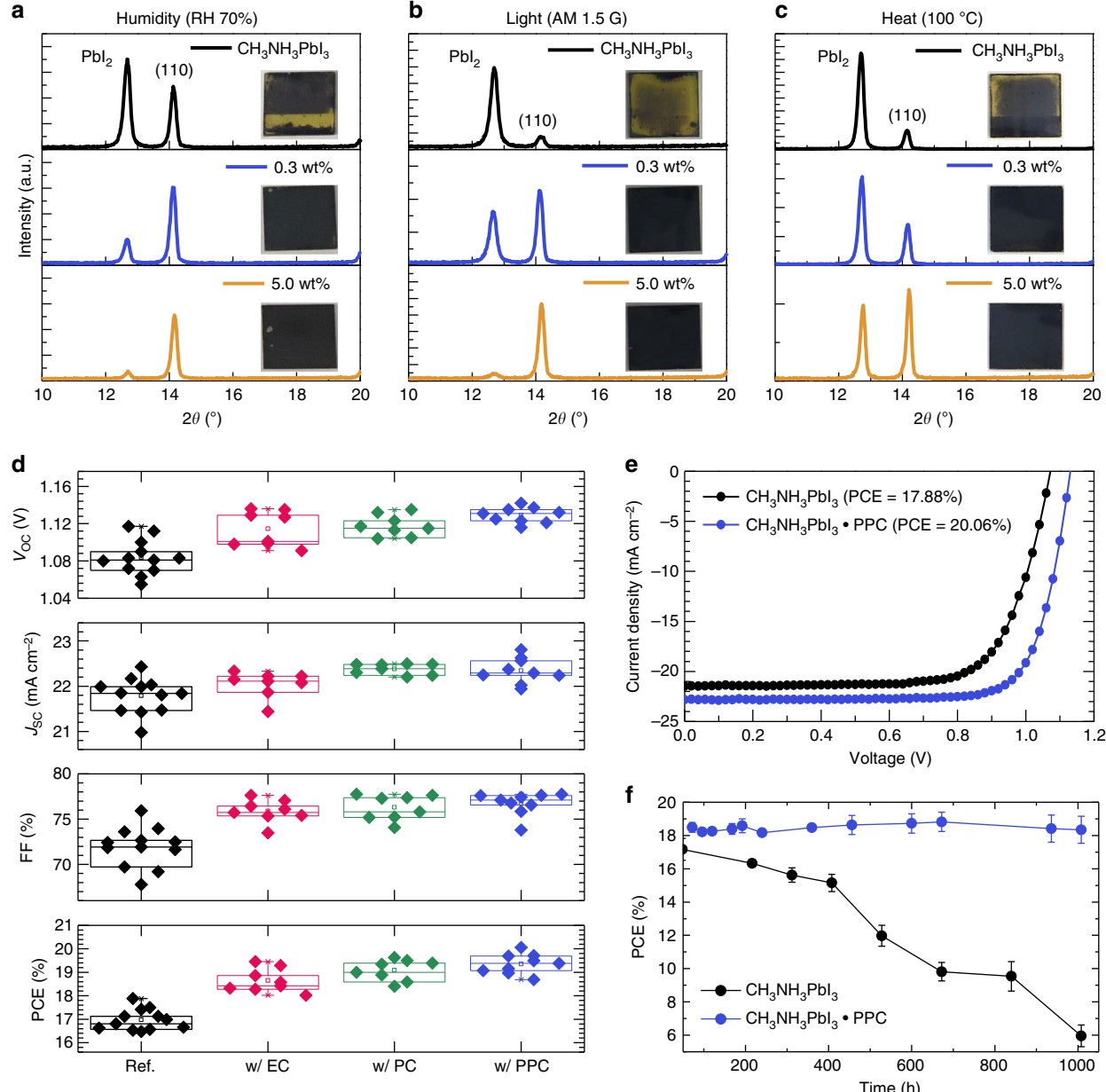

**Fig. 5** Environmental stability and photovoltaic performances. X-ray diffraction spectra of bare $CH_3NH_3PbI_3$ and $CH_3NH_3PbI_3$:PPC (0.3 and 5.0 wt%) films against **a** moisture (relative humidity: 70 ± 5%), **b** light (AM 1.5G) and **c** heat (100 ºC), **d** Photovoltaic parameters of $CH_3NH_3PbI_3$ perovskite solar cells with the addition of different Lewis bases: short-circuit current density ($J_{SC}$), open-circuit voltage ($V_{OC}$), fill factor (FF) and power conversion efficiency (PCE). Each box chart includes the minimum, lower quartile (lower horizontal line), median (middle horizontal line), mean (hollow square), upper quartile (upper horizontal line), maximum, and discrete data, **e** Current density–voltage ($J$–$V$) characteristics and **f** PCE evolution in ambient conditions as a function of time of $CH_3NH_3PbI_3$-based solar cells without and with PPC. The error bars represent the standard deviation of the PCE measured from multiple devices for each condition

**Table 2 Photovoltaic parameters of solar cells of bare $CH_3NH_3PbI_3$ and $CH_3NH_3PbI_3$ with Lewis bases**

|  | $J_{SC}$ (mA cm$^{-2}$) | $V_{OC}$ (V) | FF | PCE (%) (best PCE) |
|---|---|---|---|---|
| $CH_3NH_3PbI_3$ | 21.79 ± 0.39 | 1.084 ± 0.019 | 0.719 ± 0.022 | 16.97 ± 0.44 (17.88) |
| w/EC | 22.05 ± 0.28 | 1.115 ± 0.019 | 0.759 ± 0.013 | 18.65 ± 0.51 (19.46) |
| w/PC | 22.38 ± 0.12 | 1.118 ± 0.011 | 0.763 ± 0.014 | 19.09 ± 0.44 (19.63) |
| w/PPC | 22.34 ± 0.28 | 1.129 ± 0.008 | 0.767 ± 0.013 | 19.35 ± 0.43 (20.06) |

devices with PPC were significantly elongated. Especially, the formamidinium-cesium-based solar cells with PPC decreased its initial decay to 1.8%, and after 500 h illumination, maintained 95.0% of initial PCE, from which the expected $T_{80}$ was approximately 10,800.1 h. The slower decay has been related to a permanent degradation of the perovskite layer accompanied by a chemical reaction and morphological change[51]. Therefore, the inter-grain cross-linking of the perovskite, induced by the macromolecular intermediate phase, could very well have retarded the irreversible degradation of the perovskite as well.

## Discussion

A polymeric Lewis base with high molecular dipole moment Lewis basic repeating units was employed into polycrystalline perovskite. The polymer facilitated the formation of a long-range, molecularly ordered intermediate phase via a series of Lewis base adduct formations with the perovskite precursors. The formation of such a macromolecular adduct increased the activation energy for nucleation and diffusion of the precursor molecules, such that the subsequent perovskite films were fabricated with high crystallinity and significantly enlarged grains. Remnant polymeric Lewis bases in the perovskite films also effectively passivated the defect sites at the grain boundaries with high-binding energies. Furthermore, the long-range ordering of the perovskite precursor molecules along the polymer backbone enabled inter-grain cross-linking to form a polymer-perovskite composite bridges between perovskite grains, which minimizes inter-grain electrical decoupling and greatly improved the stability of the perovskite films against harsh environmental conditions. As a result, all the photovoltaic parameters of the devices including $J_{SC}$, $V_{OC}$, and FF were enhanced, and a highest PCE of 20.06% (stabilized efficiency of 19.48%) was achieved alongside significantly improved ambient and operational stability.

## Methods

**Materials**. A total of 1 mmol of $CH_3NH_3I$ (Dyesol) and $PbI_2$ (99.999%, TCI) were dissolved in N,N-dimethylformamide (DMF, anhydrous, 99.8%, Sigma-Aldrich) to form the perovskite precursor solution. To synthesize the adduct powder with the Lewis base molecules, each of the Lewis bases were separately added into the perovskite precursor solution. Dimethylsulfoxide (DMSO, anhydrous, Sigma-Aldrich), ethylene carbonate (EC, Sigma-Aldrich), propylene carbonate (PC, Sigma-Aldrich) and poly (propylene carbonate) (PPC, $M_n$: 50,000, Sigma-Aldrich) were added as Lewis base additives. The precursor solution was added dropwise into 10 mL of diethyl ether (DE, anhydrous, >99.0%, with BHT as a stabilizer, Sigma-Aldrich). After 10 min stirring of the solution, the precipitates were dried under vacuum.

**Device fabrication**. Indium-tin-oxide (ITO) on glass substrates were cleaned by sequential ultra-sonication in detergent, acetone, and isopropanol for 15 min each. Cleaned ITO glass was treated by ultraviolet ozone (UVO) for 15 min, and then 30 mM $SnCl_2 \cdot 2H_2O$ (98%, Sigma-Aldrich) dissolved in ethanol (anhydrous, Sigma-Aldrich) solution was spin-coated. The $SnO_2$ film on ITO glass was sequentially annealed at 150 °C for 30 min and 180 °C for 60 min. A total of 1 mmol of $CH_3NH_3I$, $PbI_2$, and DMSO were dissolved in 500 mg of DMF for the perovskite precursor solution. A total of $x$ mmol of EC or PC ($x = 0, 0.2, 0.4,$ or 0.8) was added in the perovskite precursor solution. PPC was separately dissolved in DMF and added into the perovskite solution before spin coating for the polymer-mediated perovskite film. After 15 min UVO treatment, the perovskite solution was spin coated on glass/ITO/$SnO_2$. A total of 0.3 mL of DE was dropped during spin coating (after 10 s) of the perovskite solution. The transparent adduct film was annealed at 65 °C for 1 min, and then 100 °C for 30 min. A spiro-OMeTAD solution was spin coated onto the perovskite film. Spiro-OMeTAD solution was prepared with the following composition: 85.8 mg of spiro-MeOTAD (1-Material), 33.8 μl of 4-tert-butylpyridine (96%, Sigma-Aldrich), and 19.3 μl of Li-TFSI (99.95%, Sigma-Aldrich, 520 mg mL$^{-1}$ in acetonitrile) in 1 mL of chlorobenzene (anhydrous, 99.8%, Sigma-Aldrich). For the top electrode, 80 nm-thick Ag was thermally deposited at an evaporation rate of 0.5 A s$^{-1}$.

**Computation**. Spin-polarized DFT calculations were performed using the generalized gradient approximation with the Perdew–Burke–Ernzerhof (PBE)[52] exchange correlation functional. All DFT calculations were performed using the Vienna ab-initio Simulation Package (VASP)[53]. Core electrons were treated with

the projector augmented wave (PAW) method. The cutoff energy of plane-wave expansions was set to 400 eV for all DFT calculations. DFT-D2 method of Grimme[54] was employed to treat the van der Waals interactions. For the calculation of the intermolecular interactions and the single molecules, a fermi level smearing scheme with the width of 0.01 eV and 1*1*1 gamma K-point were adopted. For the surface calculations, the fermi level smearing scheme with the width of 0.1 eV and 8*8*1 Monkhorst-pack scheme[55] were used. The 2*2*4 slab of $CH_3NH_3PbI_3$ with 15 Å vacuum layer was constructed for the surface calculations. The convergence criteria for electronic wave function and local minima were $1 \times 10^{-5}$ eV and $1 \times 10^{-2}$ eV energy differences, respectively. Bader charge analysis[56] was performed for the partial atomic charge and charge transfer between molecules and slabs. The isosurface of total charge densities were visualized with the Visualization for Electronic and Structure Analysis (VESTA)[57] software.

**Characterizations**. Fourier transform infrared (FTIR) spectroscopy was carried out by using a FT/IR-6100 (Jasco) purged by nitrogen gas. For the FTIR measurement, a sodium chloride substrate was used. Atomic force microscopy (AFM, Bruker dimension Fast Scan) with peakforce tapping mode using a 1 ohm silicon tip (OTESPA, Bruker), and scanning electron microscopy (SEM, Nova Nano 230) was used to characterize the morphology of the perovskite films. X-ray diffraction was measured by a X-ray diffractometer (PANalytical) with Cu Kα radiation at a scan rate of 4° min$^{-1}$. Photoluminescence (PL) was measured by a Horiba Jobin Yvon. Monochromatic laser (λ: 640 nm) was used for the excitation of perovskite films. Time-resolved PL was recorded by using a Picoharp 300, and a picosecond laser diode head (PLD 800B, PicoQuant) was used (λ: 640 nm, frequency: 100 kHz). Measurement of solar cells was carried out in an ambient atmosphere without pre-conditioning such as voltage bias and light soaking. Current–voltage ($J$–$V$) characterizations of the solar cells were measured with source meter, Keithley 2401, under simulated one-sun illumination (AM 1.5G, 100 mW cm$^{-2}$) (Oriel Sol3A with class AAA solar simulator (Newport)). The intensity calibration of the light was done by NREL-certified Si photodiode with a KG-5 filter. A 0.100 cm$^2$ sized metal aperture was also used onto the device to precisely define the active area during the measurement (device active area: 0.13 cm$^2$), and a scan rate was 0.1 V s$^{-1}$ (−0.1 V to 1.2 V) for $J$–$V$ characterizations. Steady-state power output of the solar cells was calculated from the photocurrent measured under constant bias voltage operation that corresponds to a maximum power. The external quantum efficiency (EQE) measurement was carried out by using specially designed system (Enli tech) under AC mode (chopping frequency: 133 Hz) without bias light.

## Data availability

The authors declare that the data supporting the findings of this study are available within the paper and its supplementary information files.

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

## Acknowledgements

This work was supported by the Air Force Office of Scientific Research (AFOSR, Grant No. FA9550-15-1-0333), Office of Naval Research (ONR, Grant No. N00014-17-1-2484), National Science Foundation (NSF, Grant No. ECCS-EPMD-1509955), and Horizon PV.

## Author contributions

T.H.H. designed and conducted most of experiments, analyzed all the data and prepared the manuscript. J.W.L. assisted in data analysis and preparation of the manuscript. S.T., Y.Z., Z.D., N.D.M., and S.H.B. assisted in experiments. C.C., S.J.L. and Y.H. conducted microstructure analysis. C.L. and H.M.L. performed density of theory calculations. Y.Y. (Yuan) assisted in stability analysis. Y.Y. (Yang) initiated the study, analyzed all the data. All authors discussed the results and contributed to the paper.

## Additional information

**Competing interests:** The authors declare no competing interests.

