## [Peer Review File · Nature Communications]

Reviewers' comments:

Reviewer #1 (Remarks to the Author):

Han et al. report macromolecular intermediate phase induced inter-grain cross-linking of organolead halide perovskites, and give a more than 20% PCE and enhanced stability. Long chain polymer molecules have been applied to cross-link grain boundaries in perovskite film to improve stability against moisture and oxygen in many previous reports. The novelty and significance of this work is less important considering many similar existed papers. This manuscript would be more appropriate for a more specified journal, such as AM or AEM.

1. The authors mentioned the problems encountered with incorporating small molecules in device stability under electric field and strong light. However, the stability under light illumination only (not real operational stability) of the device presented here shows only 50% maintenance within 500 hours, which is far from a good stability as well.
2. Long chain polymer molecules have been applied to cross-link grain boundaries in perovskite film to improve stability against moisture and oxygen in many previous reports. Nevertheless, they are missing here and even not mentioned in introduction.
3. Although the characterization is well done with rather reasonable explanation, the resultant efficiency and stability of the solar cell devices just reveal a good above average data, which can not meet the high requirements of Nature Communications.
4. Although this work reveals an improved stability in ambient conditions, but nowadays shelf stability is not enough for applications, considering that strict encapsulation would help a lot in this sense. On the other hand, operational stability is more important in terms of commercialization since it demonstrates the real working state under continuous light illumination under MPP tracking, while the operational stability of this work does not show a top-level data.
5. On page 11, the authors stated "In contrast, the polymer-perovskite crystal composite interconnector reported here could contribute electrical coupling between the polycrystalline perovskite grains." Any evidences?

Reviewer #2 (Remarks to the Author):

The paper should be revised by responding to the attached report.

Metal halide perovskite films are conventionally prepared with solution processes and subsequent crystallization. The paper describes the use of long polymeric chains, brought at early state of the processing, to increase the environmental stability and optoelectronic properties of polycrystalline perovskite solar cells. The long polymeric chains act to crosslink adjacent grains, adding to improved stability of the perovskite structure. Another impact of the polymer is to decrease the number of nucleation sites during crystallization, thereby leading to increased grain size and higher crystallinity which further improves the cell performance. The effect of long polymeric chains (polypropylene carbonate, PPC) vs (small) un-polarized molecules (ethylene carbonate, EC and propylene carbonate, PC) was investigated. These materials were chosen on the bases of their high dipole moment.

The work provides an interesting way to lock the structure and affect the grain growth by using long polymers, aiming for improved stability and performance. The analyses and arguments provided were also sound. However, the practical aspect of the finding should be addressed better.

1. The authors are mainly looking into the effect of using PCC vs the effect of using EC and PC instead. A more detailed comparison to the performance of perovskites made by other groups should be given.
2. The performance (J_{sc} , V_{oc} , FF, PCE) of the as-prepared cells is compared in Fig. 5. No description of the reference sample in Fig.5d was given. At least I did not find it.
3. One of the major reasons for the degradation of these perovskite solar cells is the migration of the Iodine. This has been shown in several publications, for example *Journal of Physical Chemistry Letters* 7, 5168 (2016), which should be cited. How the polymer can prevent the Iodine migration? The authors seem to claim that the migration of ions is reversible, but I am not sure. Can the authors clarify this aspect?
4. The PCE reported in Fig.5d seem to me quite high, or too optimistic. One typical problem in measuring the PCE of such solar cells is their long relaxation times leading to a hysteresis of the J-V characteristic. (One reason is the migration of Iodine, but also other polarization mechanisms.) Many groups overestimated the PCE because their measurement was done with too short time steps and thus was influenced by transient currents. See for example *Solar Energy Materials and Solar Cells* 159, 197 (2017) and *Solar Energy* 173, 976 (2018). Can the authors show in the J-V characteristics of Fig.5f both scan directions of the voltage, forward and reverse? Is the hysteresis area reduced by using the polymer?
5. Environmental stress test was executed by placing the sample in 70% humidity for 150 h and at elevated temperature (100°C) in inert ambient (N_2) for 66 h and exposure to light (1.5 M_0) for 2 h. The performance of the cells after the environmental test was not given. It was only shown that the PCC suffered from less degradation than EC and PC (suppl. Fig. S17). Even though that does indicate that PCC will work

better than EC and PC, it says little about the absolute performance of the cell. While these tests may be suitable to demonstrate a difference in using the different additives, a better comparison to a reference cell is needed.

Reviewer #3 (Remarks to the Author):

This work describes polymer-perovskite composite film for perovskite solar cell. The methodology is basically based on Lewis acid-base adduct approach. Poly(propylene carbonate) (PPC) was used for this study. DMSO has been usually used to form adduct but DMSO is finally removed to form perovskite. However, in this work PPC as a Lewis base was not be removed from perovskite but included in perovskite film. By doing so perovskite grain size was increased and stability was also significantly improved. Polymer-perovskite compisite approach is expected to be platform technology in perovskite solar cell research. Thus this work should be published without delay but after minor revision.

1. The title is not attractive. I suggest the following title. "Perovskite-polymer composite approach for long-term stable and high efficiency perovskite solar cell"
2. It is required to describe J-V hysteresis for the perovskite-polymer composite based solar cell.
3. Except for PPC, it is necessary to suggest other candidates (hydrophobic polymers) for perovskite-polymer composite approach.
4. Since spiro-MeOTAD was used, thermal stability may have problem?

Response to the reviewers' comments

We appreciate the reviewers' valuable comments. We revised our manuscript to comply with all of them, and have prepared point-by-point responses, which we present here. The revised parts in the manuscript are marked in red.

Reviewer #1

Comments:

Han et al. report macromolecular intermediate phase induced inter-grain cross-linking of organolead halide perovskites, and give a more than 20% PCE and enhanced stability. Long chain polymer molecules have been applied to cross-link grain boundaries in perovskite film to improve stability against moisture and oxygen in many previous reports. The novelty and significance of this work is less important considering many similar existed papers. This manuscript would be more appropriate for a more specified journal, such as AM or AEM.

1. The authors mentioned the problems encountered with incorporating small molecules in device stability under electric field and strong light. However, the stability under light illumination only (not real operational stability) of the device presented here shows only 50% maintenance within 500 hours, which is far from a good stability as well.

3. Although the characterization is well done with rather reasonable explanation, the resultant efficiency and stability of the solar cell devices just reveal a good above average data, which can not meet the high requirements of Nature Communications.

4. Although this work reveals an improved stability in ambient conditions, but nowadays shelf stability is not enough for applications, considering that strict encapsulation would help a lot in this sense. On the other hand, operational stability is more important in terms of commercialization since it demonstrates the real working state under continuous light illumination under MPP tracking, while the operational stability of this work does not show a top-level data.

Answer for (1), (3), and (4): We appreciate the reviewer's constructive comments. We have additionally fabricated perovskite solar cells with different kinds of 'A'-site cations (e.g., methylammonium (MA), formamidinium (FA) and cesium (Cs)) to demonstrate the universality of the polymeric Lewis base assisted crystallization to enhance the device stability as the reviewer pointed out. The volatile nature of MA itself causes it to easily decompose into CH₃I and NH₃ at low temperatures of ~80 °C [*Energy Environ. Sci.* **9**, 3406 (2016)], which accelerates the operational instability of the perovskite solar cells based on MAPbI₃. Also, the MA cation has 1) a lower activation energy for ion migration due to its weaker hydrogen bonding with the surrounding PbI₆ octahedra [*Nano Lett.* **14**, 3608 (2014)], 2) a smaller ionic radius and 3) a higher molecular dipole compared to FA (MA: 2.53 Å, and 2.29 D, FA: 2.53 Å, and 0.21 D) [*J. Phys. Chem. C* **121**, 14517 (2017)], which are the culprits for its operational instability. By measuring the thermal conductance of the perovskite films without/ with the polymeric Lewis base, we investigated the activation energy for ion migration in MAPbI₃. The Nernst-Einstein relation ($\sigma(T) = \frac{\sigma_0}{T} \exp\left(\frac{-E_a}{kT}\right)$, where k is the Boltzmann constant and σ_0 is a constant) was used to calculate the activation energy for ion migration, and the lateral conduction configuration was employed for the measurement (**Fig. R1 inset**). Our macromolecular adduct approach, which resultantly modified the perovskite crystallization,

effectively increased the ion migration activation energy from 0.40 eV (bare MAPbI₃) to 0.53 eV (MAPbI₃ with PPC) (**Fig. R1**), which can be attributed to the reduction of the perovskite's charged defects (e.g., positively charged anion vacancies) at the grain boundaries due to the passivation effect of the polymeric Lewis bases that remained in between the perovskite grains (**Fig. R2**).

Figure R1. Temperature-dependent conductivity of MAPbI₃ films **a**, without and **b**, with PPC (inset: the schematic illustration of the lateral conduction device configuration.)

Figure R2. a, Photoluminescence (PL) (Inset: normalized PL spectra) and **b**, time-resolved PL spectra of perovskite films without and with Lewis bases (Inset: PL lifetimes fitted from the time-resolved PL spectra).

To compare the operational stability of the perovskite solar cells with different compositions and polymeric Lewis bases, we fabricated three different kinds of perovskite solar cells, 1) MAPbI₃, 2) MA_{0.9}FA_{0.1}PbI₃, and 3) FA_{0.98}CS_{0.02}PbI₃ with 1.67 mol% of PEA₂PbI₄, with/ or

without PPC. Steady-state power conversion efficiencies (PCEs) of encapsulated devices were periodically measured with aging under continuous illumination (0.9 sun , $90 \pm 5 \text{ mW/cm}^2$), and open-circuit condition. Owing to the limited availability of our solar simulator, we unavoidably aged the devices under open-circuit condition, but it has been studied that continuous illumination under the open-circuit condition usually brings about more severe degradation of the perovskite solar cells than that with maximum power point tracking, which is attributed to the non-extracted and populated photo-generated carriers within the device, resulting in an accumulated charge driven degradation of the material [*Nat. Energy* **3**, 61 (2018)]. The PPC effectively elongated all the operational stability of the perovskite solar cells regardless of the 'A'-site cation composition. As we expected, the solar cell based on MAPbI_3 showed the fastest decay in their steady-state PCEs as compared to the other compositions due to its inherent material instability (**Fig. R3a**). When 10% of MA was replaced with FA (i.e., $\text{MA}_{0.9}\text{FA}_{0.1}\text{PbI}_3$), the operational stability was improved relative to that of bare MAPbI_3 , and the FACs-based perovskite with a small amount of 2D perovskite (1.67% of PEA_2PbI_4) further increased the operational stability. The device with the FACs-based perovskite maintained $\sim 80\%$ of initial PCE after aging for 500 h, while MAPbI_3 degraded to $\sim 20\%$, which can be attributed to the better material stability against light/ high temperature and the higher tolerance for ion migration relative to that of MAPbI_3 . All the perovskite solar cells fabricated without PPC showed an exponential initial decay, followed by linear, stabilized decay. The devices with PPC substantially reduced the amount of this initial decrease (**Fig. R4**, and **Table R1**), which could be attributed to a reduced charged defect formation, effective defect passivation, and higher activation energy for their migration with PPC. We also extracted the T_{80} (time with which the PCE decays to 80% of its initial PCE) from the post-burn-in regime (linear decay regime) for all the solar cells fabricated in this work and summarized the data in Table R1. The FACs-based solar cells with PPC decreased its initial decay to 1.8%, and after 500h

illumination, maintained 95.0% of initial PCE, from which the expected T_{80} is ~ 10800.1 h. We believe that this operational stability study demonstrates 1) the universality of our macromolecular intermediate phase approach for all organic cation based perovskites, and also that 2) our method can attain long-term operational stability if coupled with suitable compositional engineering.

Figure R3. PCE evolution under light illumination (1.5 AM) as a function of time of encapsulated solar cells without and with the polymeric Lewis base (PPC).

Figure R4. PCE evolution under light illumination (1.5 AM) as a function of time of encapsulated solar cells without and with the polymeric Lewis base (PPC).

Table R1. Initial decay and calculated T_{80} of solar cells under light illumination.

	Initial decay [%]	T_{80} [h]
MA	44.7	169.5
MA w/ PPC	29.5	391.0
MAFA	36.0	288.0
MAFA w/ PPC	16.2	929.6
FACs	10.4	420.8
FACs w/ PPC	1.8	10800.1

Revised parts in the manuscript)

Page 14-15 in the manuscript,

“To demonstrate the universality of our cross-linking approach, we examined the operational behavior of the devices based on various kinds of ‘A’-site cations (MA, MA_{0.9}FA_{0.1}, and FA_{0.98}CS_{0.02}). To evaluate their operational stability, all the devices were encapsulated under a nitrogen atmosphere and exposed to continuous illumination (90 ± 10 mW, without UV filter) under open-circuit condition (Supplementary Fig. 26, 27, and Supplementary Table 4). PPC effectively elongated the operational lifetime of all the perovskite solar cells regardless of their ‘A’-site cation composition. All the devices without and with PPC showed rapid initial decay followed by slower degradation with an almost linear profile, but the devices with PPC demonstrated a less severe initial decay. Given that the initial rapid decay can be related to the migration of ion and charged defects⁴⁸, the reduced initial decay regime can be attributed to a

decreased charged defect density and an increased activation energy for ion migration as a result of the PPC-induced crystallization. Severe ion migration in perovskite solar cells can result in J-V hysteresis, and along with the possible compound formation with the electrode, could accelerate the degradation of a device during operation⁴⁹⁻⁵¹. Indeed, PPC effectively increased the activation energy for ion migration in the perovskite (Supplementary Fig. 28), resulting in reduced hysteresis than that of bare $\text{CH}_3\text{NH}_3\text{PbI}_3$ (Supplementary Fig. 29). Furthermore, the addition of PPC slowed down the subsequent linear decay regime as well. From this, we extracted the T_{80} (time with which the PCE decays to 80% of its initial value) from the post-burn-in regime (linear decay regime) for all the solar cells fabricated in this work and summarized this in Supplementary Table S4. All the calculated T_{80} lifetimes of the devices with PPC were significantly elongated. Especially, the FACs-based solar cells with PPC decreased its initial decay to 1.8%, and after 500h illumination, maintained 95.0% of initial PCE, from which the expected T_{80} is ~ 10800.1 h. The slower decay has been related to a permanent degradation of the perovskite layer accompanied by a chemical reaction and morphological change⁵². Therefore, the inter-grain cross-linking of the perovskite, induced by the macromolecular intermediate phase, could very well have retarded the irreversible degradation of the perovskite as well.”

Figure S26. Operational stability of solar cells. PCE evolution under light illumination (1.5 AM) as a function of time of encapsulated solar cells without and with the polymeric Lewis base (PPC).

Figure S27. Operational stability of solar cells. PCE evolution under light illumination (1.5 AM) as a function of time of encapsulated solar cells without and with the polymeric Lewis base (PPC).

Table S4. Initial decay and calculated T₈₀ of solar cells under light illumination.

	Initial decay [%]	T ₈₀ [h]
MA	44.7	169.5
MA w/ PPC	29.5	391.0
MAFA	36.0	288.0
MAFA w/ PPC	16.2	929.6
FACs	10.4	420.8
FACs w/ PPC	1.8	10800.1

The Nernst-Einstein relation ($\sigma(T) = \frac{\sigma_0}{T} \exp(\frac{-E_a}{kT})$, where k is the Boltzmann constant and σ_0 is a constant) was used to calculate the activation energy for ion migration, and the lateral conduction configuration was employed for the measurement (Fig. S28 inset). Our macromolecular adduct approach, which resultantly modified the perovskite crystallization, effectively increased the ion migration activation energy from 0.40 eV (bare CH₃NH₃PbI₃) to 0.53 eV (CH₃NH₃PbI₃ with PPC) (Fig S28), which can be attributed to the reduction of the perovskite's charged defects (e.g., positively charged anion vacancies) at the grain boundaries due to the passivation effect of the polymeric Lewis bases that remained in between the perovskite grains, and this translated to a reduced J-V hysteresis of the solar cells with PPC.

Figure S28. Activation energy for ion migration. Temperature-dependent conductivity of MAPbI₃ film **a**, without and **b**, with PPC (inset: schematic illustration of the lateral conduction device configuration.)

2. Long chain polymer molecules have been applied to cross-link grain boundaries in perovskite film to improve stability against moisture and oxygen in many previous reports. Nevertheless, they are missing here and even not mentioned in introduction.

Answer: As the reviewer pointed out, we additionally cited 8 more previous literatures regarding polymeric additives [*ACS Appl. Mater. Interfaces* **7**, 4955 (2015), *Nat. Commun.* **7**, 10228 (2016), *Nat. Energy* **1**, 16142 (2016), *Adv. Energy Mater.* **8**, 1701757 (2018), *Chem* **4**, 1404 (2018), *Energy Environ. Sci.* **11**, 2609 (2018)] or crosslinking approaches using small-molecular crosslinking agents [*Nat. Commun.* **9**, 3806 (2018)], and also mentioned previous studies regarding such polymer additives and perovskite cross-linking. We appreciate the reviewer's helpful comment.

Revised parts in the manuscript)

Page 4-5 in the manuscript,

“the high degree of volatility and high diffusion coefficients of small molecules likely pose difficulties in incorporating them into practical devices operated under harsh environments such as high temperature, humidity, electric field and strong light. For these reasons above, there have been many attempts to use polymeric additives^{21, 31-37} or cross-linking small molecular agents^{38,39} incorporated into the perovskite active layer as a crystal growth template, defect passivating molecule or crystal cross-linker. Compared to small molecules, polymeric molecules are expected to provide high molecular stability and polymer-assisted cross-linking of the crystal grains of perovskite which improve the morphological and environmental stability of the perovskite films. Furthermore, long-chain polymers could be immobilized after crystallization of the perovskite, while small molecule additives have substantial diffusion and drift mobility in the perovskite film during operation. Here, we introduce a new type of inter-grain cross-linking, induced by a macromolecular intermediate phase with a polymeric Lewis base, for one-step fabrication of highly efficient and stable perovskite solar cells. Repeating units of the long-chained polymeric Lewis base can possibly form Lewis acid-base adducts with a series of perovskite precursor molecules, so it forms a macromolecular intermediate phase with a much larger degree of coordination and long-range molecular ordering along the polymer chain (Fig. 1b). This subsequently leads to the formation of enlarged, cross-linked perovskite grains, which are also defect-passivated by the remnant polymeric Lewis bases. Consequently, it is possible to realize highly efficient and stable perovskite solar cells.”

Page 19-20 in the manuscript,

32. Chang, C.-Y. *et al.* Tuning perovskite morphology by polymer additive for high efficiency solar cell. *ACS Appl. Mater. Interfaces* **7**, 4955-4961 (2015).
33. Zhao, Y. *et al.* A polymer scaffold for self-healing perovskite solar cells. *Nat. Commun.* **7**, 10228 (2016).
34. Bi, D. *et al.* Polymer-templated nucleation and crystal growth of perovskite films for solar cells with efficiency greater than 21%. *Nat. Energy* **1**, 16142 (2016).
35. Jiang, J. *et al.* Polymer doping for high-efficiency perovskite solar cells with improved moisture stability. *Adv. Energy Mater.* **8**, 1701757 (2018).
36. Zong, Y. Continuous grain-boundary functionalization for high-efficiency perovskite solar cells with exceptional stability. *Chem* **4**, 1404-1415 (2018).
37. Kim, M. *et al.* Enhanced solar cell stability by hygroscopic polymer passivation of metal halide perovskite thin film. *Energy Environ. Sci.* **11**, 2609-2619 (2018).
38. Li, X. *et al.* Improved performance and stability of perovskite solar cells by crystal crosslinking with alkylphosphonic acid ω -ammonium chlorides. *Nat. Chem.* **7**, 703-711 (2015).
39. Li, X. *et al.* In-situ cross-linking strategy for efficient and operationally stable methylammonium lead iodide solar cells *Nat. Commun.* **9**, 3806 (2018).

5. On page 11, the authors stated “In contrast, the polymer-perovskite crystal composite interconnector reported here could contribute electrical coupling between the polycrystalline perovskite grains.” Any evidences?

Answer: We have additionally compared perovskite solar cells incorporated with three different additives with different molecular structures and functional groups to examine their effect on the electrical properties of the perovskite polycrystals. We used a small molecular Lewis base (i.e., propylene carbonate (PC)), a polymeric Lewis base (i.e., PPC), and a polymeric acid (i.e., polyacrylic acid (PAA)). The repeating functional group of PAA is the simplest unsaturated carboxylic acid (C_2H_3COOH). We explored the photovoltaic performances of the perovskite solar cells according to different additive concentrations, ranging from 0.1wt% to 5.0wt%. The small molecule (PC) and polymeric Lewis base (PPC) addition mainly improved the open circuit voltages (V_{OC}) and fill factors (FF) of the devices, and showed no noticeable decrease even at high concentrations (**Fig. R5**).

Figure R5. Photovoltaic parameters of $MAPbI_3$ perovskite solar cells with PC, PPC, and PAA according to their molecular concentrations. (short-circuit current density (J_{SC}), open-circuit voltage (V_{OC}), fill factor (FF) and power conversion efficiency (PCE)).

When 0.3wt% of PPC (optimum concentration in this work) was added, all the parameters of the MAPbI₃-based solar cells increased (V_{OC} : 1.084±0.018 to 1.129±0.008 V, J_{SC} : 21.79±0.39 to 22.35±0.28 mA/cm², and FF: 71.9±2.2 to 76.7±1.3%), resulting in a highly improved PCE (17.0±0.5% to 19.4±0.4%). Even with the addition of 5.0wt% (corresponding to ~17 times higher than the optimum amount), the devices with PPC did not exhibit a severe decrease in performance (V_{OC} : 1.137±0.009 V, J_{SC} : 20.83±0.25 mA/cm², FF: 75.2±1.5%, and PCE: 17.8±0.4%) and were still superior than those of bare MAPbI₃. In contrast, as the molecular concentration of PAA in the perovskite precursor solution increases, all the photovoltaic parameters gradually decreased. Particularly, the FF and J_{SC} of the devices showed the most obvious decaying trends as more PAA was added (FF: 71.9±2.2% to 17.3 ±1.8%, and J_{SC} : 21.8±0.4 mA/cm² to 13.2±1.7 mA/cm²), which clearly indicates that the electrical conduction and charge extraction became worse with more PAA in the perovskite film. As a result, with a high amount of PAA added, the PCE substantially degraded to 5.5±1.4% as compared to the bare MAPbI₃ devices (PCE: 17.0±0.4%) (**Fig. R5**).

The different influences of the polymers on crystal growth and electrical conduction were examined by analyzing the surface morphology and spatially resolved electrical conductivity of the perovskite films. As we reported in the manuscript, PPC has a strong chemical interaction with the perovskite precursors (i.e., acid-base adduct formation) (Supplementary Fig. S1-4, and Table S2), forming a macromolecular long-range intermediate phase which increases the activation energy for crystal growth, consequently increasing the grain size of the perovskite crystals (Supplementary Fig. S6, and S7). On the contrary, we found that PAA has the opposite effect on perovskite crystal growth compared to that of PPC. PAA gradually decreased the

grain size as higher amounts of PAA was added (**Fig. R6**). All the MAPbI₃ grains were finely cleaved into nanometer-sized grains upon the addition of 5.0wt% of PAA (**Fig. R6c**), which could be attributed to the chemical interaction between PAA and the perovskite precursors/ or solvent in the solution.

Figure R6. Atomic force microscopy images of MAPbI₃ film **a**, without, **b**, with 0.3wt% of PAA, and **c**, with 5.0wt% of PAA.

Per classical theory for homogeneous nucleation, the nucleation rate is described by using a critical free energy (ΔG_c), which represents the free energy required for nuclei to be stable without being dissolved in the solution and is the sum of the surface and bulk free energy of the nuclei. This critical free energy is defined as the activation energy for nucleation and used to describe the nucleation rate using an Arrhenius type equation [*Chem. Rev.* **114**, 7610 (2014), *Small* **7**, 2685 (2011)]:

$$\frac{dN}{dt} = A \exp\left(-\frac{\Delta G_c}{k_B T}\right) \quad (1)$$

where t is time, N is number of nuclei, A is pre-exponential factor, k_B is Boltzmann's constant, T is temperature. The crystal free energy (ΔG_c) can be written as a function of surface energy γ , molar volume v , supersaturation of solution S , which produce a following equation,

$$\frac{dN}{dt} = A \exp\left(-\frac{16\pi\gamma^3 v^2}{3k_B^3 T^3 (\ln S)^2}\right)$$

Because the repeating functional group of PAA is a carboxylic acid (C_2H_3COOH) ($pK_a=4.25$), the PAA likely interacted with the Lewis base solvents of the precursor solution (i.e., DMSO, and DMF), instead of forming an adduct with the Lewis acidic precursors, which possibly increased the saturation level of the precursors in the solution, resulting in a decreased activation energy for crystallization (**Fig. R7**). As a result, the fast crystallization formed a large number of nuclei, and subsequently small crystal grains, which could severely interrupt the inter-grain electrical coupling due to the insulating nature of PAA residing in between the small perovskite polycrystals.

Figure R7. Schematic diagrams of free energy vs. cluster radius.

The reduction of crystal grain size and electrical insulation can also be considered in the aspect of inhibitor molecules by the following model [*Theor. Found. Chem. Eng.* **42**, 179 (2008)],

$$w(C) \sim \begin{cases} w(0) \exp(-K_I C), & C < C_C \\ 0, & C > C_C \end{cases} \quad K_I = \frac{4Kf^{\frac{1}{2}}\gamma V_m}{kT\sigma\pi^{\frac{1}{2}}}$$

where w is crystallization rate [m/s], C is the inhibitor concentration [m^{-3}], C_C is the critical value of C , K is the adsorption constant [m], K_I is the coefficient of inhibition effectiveness

[m^3], f is the area occupied by the adsorbed impurity particle [m^2], γ is the growth step energy [J m^{-2}], V_m is the unit volume [m^3] of the crystal, and σ is the relative solution supersaturation [dimensionless]. The equation quantifies the influence of inhibitors on the crystallization rate. If the PAA does not have meaningful chemical interaction with the precursors, this additive can be regarded as an inhibitor for crystal growth. As C increases, the crystallization rate exponentially decreases, thereby resulting in small grain sizes of crystals with inhibitor molecules in between the crystals.

To directly compare and visualize the electrical properties of the perovskite grains, we performed spatially-resolved electrical conductivity measurements (conductive AFM) on the perovskite films grown without/ with PC, PPC, and PAA (**Fig. R8**). The perovskite films were deposited on an ITO/ SnO_2 electron transporting layer. To analyze their charge carrier conducting characteristics, a positive bias was applied using a Sb-doped Si tip. Both the small molecular Lewis base and polymeric Lewis base resulted in the overall electrical enhancement of the perovskite films compared to that of the bare MAPbI_3 film. Particularly, the electrical conductivity in the grain boundaries was seen to significantly increase even though the film had high concentrations of the insulating organic molecules (5.0wt%). The conductivity enhancements could be attributed to an enhanced charge carrier mobility due to the higher crystallinity and an increased charge carrier concentration due to a reduction in both the defect trapping sites and non-radiative recombination in the perovskite grains. In contrast, the MAPbI_3 film with 5.0wt% PAA incorporated exhibited electrically decoupled tiny grains and inhomogeneity in between the perovskite grains (**Fig. R8d**). As a result, the existence of the insulating PAA in the perovskite film severely disrupted the electrical properties of the solar cells, degrading its J_{SC} and FF significantly (**Fig. R8e**).

Figure R8. Electrical current mapping measured by conductive atomic force microscopy of MAPbI₃ **a**, without, and with **b**, 5.0wt% of PC, **c**, 5.0wt% of PPC, **d**, 5.0wt% of PAA. **e**, current density vs. voltage (J-V) characteristics of perovskite solar cell without, and with 5.0wt% of PC, PPC, and PAA.

Unlike small molecular additives, polymers have higher molecular length and lower molecular chain mobility, which seems to be important factors to sustain suitable interactions with the perovskite precursors and mediate the perovskite crystallization. We found that PPC, with a strong electron donating nature, strongly interacted with the perovskite precursors to form a long-range macromolecular intermediate phase. The increased activation energy for crystallization enabled enlarged crystal grains to form and the low long chain mobility of the macromolecular intermediate phase allowed the polycrystals to be crosslinked with polymer-perovskite crystal composite bridges (**Fig. 3**). For the reason above, PPC and PAA had obviously different influences on the crystal growth and electrical properties of the perovskite films. Therefore, we believe that our inter-grain crosslinking method which involves a macromolecular polymer-perovskite composite did not lead to a severe electrical degradation of the perovskite, even when incorporated with high concentrations of the insulating additive

molecules. Simultaneously, these perovskite films possessed a combination of advantageous optoelectronic/ electrical properties and material/ device stability. According to the reviewer's comment, we revised our discussion regarding the electrical coupling to be "Conventional small molecular chemical cross-linkers or conventional passivating molecules likely form secondary pure organic phases between the perovskite grains³⁶, and these organic phases do not absorb light and can possibly electrically decouple the perovskite grains. In contrast, the polymer-perovskite crystal composite interconnector reported here could minimize the electrical decoupling or insulation between the perovskite grains due to its unique composition."

Revised parts in the manuscript)

Page 9-11 in the manuscript,

"Conventional passivating molecules likely form secondary pure organic phases between the perovskite grains^{20,36}, and these organic phases do not absorb light and can possibly electrically decouple the perovskite grains. In contrast, the polymer-perovskite crystal composite interconnector reported here could minimize the electrical decoupling or insulation between the perovskite grains due to its unique composition."

"The ability to form an appropriate interaction between the polymer and the perovskite precursors crucially affects the perovskite crystal growth and the resulting electrical properties of the perovskite film. The Lewis basicity and molecular dipole of a polymer's repeating functional group also affects the formation of the macromolecular intermediate phase with the high dipole moment perovskite precursor CH_3NH_3^+ ($\mu = 2.3\text{D}$). We compared the effects of

different polymer functional groups on the perovskite crystal growth and subsequent electrical properties (See supplementary information, Fig. S9-14) and observed that the polymeric Lewis base increased the electrical conductivity of the perovskite while in contrast, the polymeric acid severely interrupted the spatial electrical conductance of the perovskite grains, which indeed resulted in a huge drop in the photovoltaic performance of the devices with a high dose of the polymeric acid molecules.”

Page 13-17 in the supplementary information,

Figure S11. Photovoltaic performance of solar cells according to molecular concentration. Photovoltaic parameters of $\text{CH}_3\text{NH}_3\text{PbI}_3$ perovskite solar cells with PC, PPC, and PAA

according to their molecular concentration. (short-circuit current density (J_{sc}), open-circuit voltage (V_{oc}), fill factor (FF) and power conversion efficiency (PCE)).

The different effects of the small molecular Lewis base (i.e., propylene carbonate (PC)), polymeric Lewis base (i.e., PPC), and polymeric acid (i.e., polyacrylic acid (PAA)) on the electrical properties of the films and subsequent photovoltaic performances of the devices were investigated. We examined the photovoltaic performances of the perovskite solar cells according to the molecular concentrations ranging from 0.1wt% to 5.0wt%. We observed that the small molecule (PC) and polymeric Lewis base (PPC) mainly improved the open circuit voltages (V_{oc}) and fill factors (FF) of the devices, which showed no noticeable decrease even at high concentrations (Fig. S11). In contrast, PAA gradually decreased the grain size with increasing amounts of PAA added (Fig. S12). All the $CH_3NH_3PbI_3$ grains were finely cleaved into nanometer-sized grains upon the addition of 5.0wt% of PAA (Fig. S12c), which could be attributed to the chemical interaction between PAA and the perovskite precursors/ or solvent in the solution.

Figure S12. Surface Morphology of perovskite films with PAA. Atomic force microscopy images of $CH_3NH_3PbI_3$ film **a**, without, **b**, with 0.3wt% of PAA, and **c**, with 5.0wt% of PAA.

Figure S13. Schematic diagrams of free energy vs. cluster radius.

Per classical theory for homogeneous nucleation, the nucleation rate is described by using a critical free energy (ΔG_c), which represents the free energy required for nuclei to be stable without being dissolved in the solution and is the sum of the surface and bulk free energy of the nuclei. This critical free energy is defined as the activation energy for nucleation and used to describe the nucleation rate using an Arrhenius type equation [*Chem. Rev.* **114**, 7610 (2014), *Small* **7**, 2685 (2011)]:

$$\frac{dN}{dt} = A \exp\left(-\frac{\Delta G_c}{k_B T}\right) \quad (1)$$

where t is time, N is number of nuclei, A is pre-exponential factor, k_B is Boltzmann's constant, T is temperature. The crystal free energy (ΔG_c) can be written as a function of surface energy γ , molar volume v , supersaturation of solution S , which produce a following equation,

$$\frac{dN}{dt} = A \exp\left(-\frac{16\pi\gamma^3 v^2}{3k_B^3 T^3 (\ln S)^2}\right)$$

Because the repeating functional group of PAA is a carboxylic acid (C_2H_3COOH) ($pK_a = 4.25$), the PAA likely interacted with the Lewis base solvents of the precursor solution (i.e., DM SO, and DMF), instead of forming an adduct with the Lewis acidic precursors, which possibly increased the saturation level of the precursors in the solution, resulting in a decreased activ

ation energy for crystallization (**Fig. S13**). As a result, the fast crystallization formed a large number of nuclei, and subsequently small crystal grains, which could severely interrupt the inter-grain electrical coupling due to the insulating nature of PAA residing in between the perovskite polycrystals.

The reduction of crystal grain size and electrical insulation can also be considered in the aspect of inhibitor molecules by the following model [*Theor. Found. Chem. Eng.* **42**, 179 (2008)],

$$w(C) \sim \begin{cases} w(0) \exp(-K_I C), & C < C_c \\ 0, & C > C_c \end{cases} \quad K_I = \frac{4Kf^{\frac{1}{2}}\gamma V_m}{kT\sigma\pi^{\frac{1}{2}}}$$

where w is crystallization rate [m/s], C is the inhibitor concentration [m^{-3}], C_c is the critical value of C , K is the adsorption constant [m], K_I is the coefficient of inhibition effectiveness [m^3], f is the area occupied by the adsorbed impurity particle [m^2], γ is the growth step energy [J m^{-2}], V_m is the unit volume [m^3] of the crystal, and σ is the relative solution supersaturation [dimensionless]. The equation quantifies the influence of inhibitors on the crystallization rate. If the PAA does not have meaningful chemical interaction with the precursors, this additive can be regarded as an inhibitor for crystal growth. As C increases, the crystallization rate exponentially decreases, thereby resulting in small grain sizes of crystals with inhibitor molecules in between the crystals.

Figure S14. Electrical properties of perovskite films. Electrical current mapping measured by conductive atomic force microscopy of CH₃NH₃PbI₃ **a**, without, and with **b**, 5.0wt% of PC, **c**, 5.0wt% of PPC, **d**, 5.0wt% of PAA. **e**, current density vs. voltage (J-V) characteristics of perovskite solar cell without, and with 5.0wt% of PC, PPC, and PAA.

The perovskite films were deposited on an ITO/ SnO₂ electron transporting layer. To analyze their charge carrier conducting characteristics, a positive bias was applied using a Sb-doped Si tip. Both the small molecular Lewis base and polymeric Lewis base resulted in the overall electrical enhancement of the perovskite films compared to that of the bare CH₃NH₃PbI₃ film. Particularly, the electrical conductivity in the grain boundaries was seen to significantly increase even though the film had high concentrations of the insulating organic molecules (5.0wt%). The conductivity enhancements could be attributed to an enhanced charge carrier mobility due to the higher crystallinity and an increased charge carrier concentration due to a reduction in both the defect trapping sites and non-radiative recombination in the perovskite grains. In contrast, the CH₃NH₃PbI₃ film with 5.0wt% PAA incorporated exhibited electrically decoupled tiny grains and inhomogeneity in between the perovskite grains (Fig. S14d). As a

result, the existence of the insulating PAA in the perovskite film severely disrupted the electrical properties of the solar cells, degrading its J_{SC} and FF significantly (Fig. S14e).

Reviewer #2

Comments:

Metal halide perovskite films are conventionally prepared with solution processes and subsequent crystallization. The paper describes the use of long polymeric chains, brought at early state of the processing, to increase the environmental stability and optoelectronic properties of polycrystalline perovskite solar cells. The long polymeric chains act to crosslink adjacent grains, adding to improved stability of the perovskite structure. Another impact of the polymer is to decrease the number of nucleation sites during crystallization, thereby leading to increased grain size and higher crystallinity which further improves the cell performance. The effect of long polymeric chains (polypropylene carbonate, PPC) vs (small) un-polarized molecules (ethylene carbonate, EC and propylene carbonate, PC) was investigated. These materials were chosen on the bases of their high dipole moment. The work provides an interesting way to lock the structure and affect the grain growth by using long polymers, aiming for improved stability and performance. The analyses and arguments provided were also sound. However, the practical aspect of the finding should be addressed better.

1. The authors are mainly looking into the effect of using PCC vs the effect of using EC and PC instead. A more detailed comparison to the performance of perovskites made by other groups should be given.

Answer: We appreciate the reviewer's constructive comments. As the reviewer suggested, we additionally compared the performances of perovskite solar cells incorporated with additives with different functional repeating groups, which can suggest new guidelines related to the functional group requirements of the polymers. We used three additional kinds of commercially

available polymers, polyacrylic acid (PAA), poly(4-vinylpyridine) (PVP), and polyurethane (PU) (**Fig. R1a**). All the polymers have lone pairs of electrons on nitrogen or oxygen along the polymer backbone, but different basicity and molecular dipole moments. Acrylic acid, the repeating unit of PAA, is the simplest unsaturated carboxylic acid (C_2H_3COOH) ($pK_a= 4.25$) with a molecular dipole moment of 1.46 D. Pyridine (C_5H_5N) and urea (CH_4N_2O) are the Lewis base functional units of PVP and PU, respectively, but the dipole moment of urea (4.56 D) is higher than that of pyridine (2.2 D). In other words, we chose three different polymers representative of a polymeric acid (PAA), a Lewis base with high (PU) and low (PVP) molecular dipole moment functional groups. As shown in Figure R1b and c, compared to the reference device ($MAPbI_3$ without polymeric additives) fabricated in the same batch, the addition of PAA and PVP into the $MAPbI_3$ perovskite degraded its photovoltaic performance, while a small amount of PU increased the photovoltaic performance of the perovskite solar cells. The addition of PU mainly improved the open circuit voltage (V_{OC}) to 1.155 V and fill factor (FF) to 71.3%, increasing the power conversion efficiency (PCE) of the $MAPbI_3$ perovskite solar cells (from 16.5% to 17.7% PCE), similar to the effect of adding PPC.

Figure R1. **a**, Chemical structure of polyacrylic acid (PAA), polyvinylpyridine (PVP), and polyurethane (PU), **b**, Current density versus voltage (J-V) characteristics of the best CH₃NH₃PbI₃ solar cells with various polymers, and **c**, photovoltaic parameters of CH₃NH₃PbI₃ perovskite solar cells with different kinds of polymers. (short-circuit current density (J_{SC}), open-circuit voltage (V_{OC}), fill factor (FF) and power conversion efficiency (PCE))

As the PU concentration increased (0.1wt% and 0.3wt%), the V_{OC} and FF increased, while the J_{SC} showed no noticeable decrease, resulting in an improved PCE (from 16.5% to 17.7%) (**Fig. R2**). In contrast, the J_{SC} and FF severely decreased (J_{SC}: 21.3 mA/cm² to 17.2 mA/cm², FF: 69.6% to 58.7%) as the PAA additive concentration in the perovskite precursor solution was increased (**Fig. R3**), which means that PAA significantly degraded the electrical properties of MAPbI₃.

Figure R2. a, photovoltaic parameters of $\text{CH}_3\text{NH}_3\text{PbI}_3$ perovskite solar cells as a function of concentration of PAA added (short-circuit current density (J_{sc}), open-circuit voltage (V_{oc}), fill factor (FF) and power conversion efficiency (PCE)), **b**, current density versus voltage (J-V) characteristics of the best $\text{CH}_3\text{NH}_3\text{PbI}_3$ solar cells with 0.1 wt% and 0.3 wt% PAA.

Figure R3. a, photovoltaic parameters of $\text{CH}_3\text{NH}_3\text{PbI}_3$ perovskite solar cells as a function of concentration of PU added (short-circuit current density (J_{sc}), open-circuit voltage (V_{oc}), fill

factor (FF) and power conversion efficiency (PCE)), **b**, current density versus voltage (J-V) characteristics of the best $\text{CH}_3\text{NH}_3\text{PbI}_3$ solar cells with 0.1wt% and 0.3wt% PU.

We have additionally compared perovskite solar cells incorporated with three different additives (i.e., propylene carbonate (PC), polypropylene carbonate (PPC), and PAA) with different molecular structures and functional groups to examine their effect on the electrical properties of the perovskite polycrystals. (**Figure R4**). We explored the photovoltaic performances of the perovskite solar cells according to different additive concentrations, ranging from 0.1wt% to 5.0wt% (**Fig. R4**). When 0.3wt% of PPC (optimum concentration in this work) was added, all the parameters of the MAPbI_3 -based solar cells increased (V_{OC} : 1.084 ± 0.018 to 1.129 ± 0.008 V, J_{SC} : 21.79 ± 0.39 to 22.35 ± 0.28 mA/cm^2 , and FF: 71.9 ± 2.2 to $76.7\pm 1.3\%$), resulting in a highly improved PCE ($17.0\pm 0.5\%$ to $19.4\pm 0.4\%$). Even with the addition of 5.0wt% (corresponding to ~17 times higher than the optimum amount), the devices with PPC did not exhibit a severe decrease in performance (V_{OC} : 1.137 ± 0.009 V, J_{SC} : 20.83 ± 0.25 mA/cm^2 , FF: $75.2\pm 1.5\%$, and PCE: $17.8\pm 0.4\%$) and were still superior than those of bare MAPbI_3 . In contrast, as the molecular concentration of PAA in the perovskite precursor solution increases, all the photovoltaic parameters gradually decreased. Particularly, the FF and J_{SC} of the devices showed the most obvious decaying trends as more PAA was added (FF: $71.9\pm 2.2\%$ to $17.3\pm 1.8\%$, and J_{SC} : 21.8 ± 0.4 mA/cm^2 to 13.2 ± 1.7 mA/cm^2), which clearly indicates that the electrical conduction and charge extraction became worse with more PAA in the perovskite film. As a result, with a high amount of PAA added, the PCE substantially degraded to $5.5\pm 1.4\%$ as compared to the bare MAPbI_3 devices (PCE: $17.0\pm 0.4\%$) (**Fig. R4**).

Figure R4. Photovoltaic parameters of MAPbI₃ perovskite solar cells with PC, PPC, and PAA according to their molecular concentration. (short-circuit current density (J_{sc}), open-circuit voltage (V_{oc}), fill factor (FF) and power conversion efficiency (PCE)).

The different influences of the polymers on crystal growth and electrical conduction were examined by analyzing the surface morphology and spatially resolved electrical conductivity of the perovskite films. As we reported in the manuscript, PPC has a strong chemical interaction with the perovskite precursors (i.e., acid-base adduct formation) (Supplementary Fig. S1-4, and Table S2), forming a macromolecular long-range intermediate phase which increases the activation energy for crystal growth, consequently increasing the grain size of the perovskite crystals (Supplementary Fig. S6, and S7). On the contrary, we found that PAA has the opposite effect on perovskite crystal growth compared to that of PPC. PAA gradually decreased the grain size as higher amounts of PAA was added (**Fig. R5**). All the MAPbI₃ grains were finely

cleaved into nanometer-sized grains upon the addition of 5.0wt% of PAA (**Fig. R5c**), which could be attributed to the chemical interaction between PAA and the perovskite precursors/ or solvent in the solution.

Figure R5. Atomic force microscopy images of MAPbI₃ film **a**, without, **b**, with 0.3wt% of PAA, and **c**, with 5.0wt% of PAA.

Per classical theory for homogeneous nucleation, the nucleation rate is described by using a critical free energy (ΔG_c), which represents the free energy required for nuclei to be stable without being dissolved in the solution and is the sum of the surface and bulk free energy of the nuclei. This critical free energy is defined as the activation energy for nucleation and used to describe the nucleation rate using an Arrhenius type equation [*Chem. Rev.* **114**, 7610 (2014), *Small* **7**, 2685 (2011)]:

$$\frac{dN}{dt} = A \exp\left(-\frac{\Delta G_c}{k_B T}\right) \quad (1)$$

where t is time, N is number of nuclei, A is pre-exponential factor, k_B is Boltzmann's constant, T is temperature. The crystal free energy (ΔG_c) can be written as a function of surface energy γ , molar volume v , supersaturation of solution S , which produce a following equation,

$$\frac{dN}{dt} = A \exp\left(-\frac{16\pi\gamma^3 v^2}{3k_B^3 T^3 (\ln S)^2}\right)$$

Because the repeating functional group of PAA is a carboxylic acid (C₂H₃COOH) (pK_a= 4.25), the PAA likely interacted with the Lewis base solvents of the precursor solution (i.e., DMSO,

and DMF), instead of forming an adduct with the Lewis acidic precursors, which possibly increased the saturation level of the precursors in the solution, resulting in a decreased activation energy for crystallization (**Fig. R6**). As a result, the fast crystallization formed a large number of nuclei, and subsequently small crystal grains, which could severely interrupt the inter-grain electrical coupling due to the insulating nature of PAA residing in between the small perovskite polycrystals.

Figure R6. Schematic diagrams of free energy vs. cluster radius.

The reduction of crystal grain size and electrical insulation can also be considered in the aspect of inhibitor molecules by the following model [*Theor. Found. Chem. Eng.* **42**, 179 (2008)],

$$w(C) \sim \begin{cases} w(0) \exp(-K_I C), & C < C_C \\ 0, & C > C_C \end{cases} \quad K_I = \frac{4Kf^{\frac{1}{2}}\gamma V_m}{kT\sigma\pi^{\frac{1}{2}}}$$

where w is crystallization rate [m/s], C is the inhibitor concentration [m^{-3}], C_C is the critical value of C , K is the adsorption constant [m], K_I is the coefficient of inhibition effectiveness [m^3], f is the area occupied by the adsorbed impurity particle [m^2], γ is the growth step energy [J m^{-2}], V_m is the unit volume [m^3] of the crystal, and σ is the relative solution supersaturation [dimensionless]. The equation quantifies the influence of inhibitors on the crystallization rate.

If the PAA does not have meaningful chemical interaction with the precursors, this additive can be regarded as an inhibitor for crystal growth. As C increases, the crystallization rate exponentially decreases, thereby resulting in small grain sizes of crystals with inhibitor molecules in between the crystals.

To directly compare and visualize the electrical properties of the perovskite grains, we performed spatially-resolved electrical conductivity measurements (conductive AFM) on the perovskite films grown without/ with PC, PPC, and PAA (**Fig. R7**). The perovskite films were deposited on an ITO/ SnO₂ electron transporting layer. To analyze their charge carrier conducting characteristics, a positive bias was applied using a Sb-doped Si tip. Both the small molecular Lewis base and polymeric Lewis base resulted in the overall electrical enhancement of the perovskite films compared to that of the bare MAPbI₃ film. Particularly, the electrical conductivity in the grain boundaries was seen to significantly increase even though the film had high concentrations of the insulating organic molecules (5.0wt%). The conductivity enhancements could be attributed to an enhanced charge carrier mobility due to the higher crystallinity and an increased charge carrier concentration due to a reduction in both the defect trapping sites and non-radiative recombination in the perovskite grains. In contrast, the MAPbI₃ film with 5.0wt% PAA incorporated exhibited electrically decoupled tiny grains and inhomogeneity in between the perovskite grains (**Fig. R7d**). As a result, the existence of the insulating PAA in the perovskite film severely disrupted the electrical properties of the solar cells, degrading its J_{SC} and FF significantly (**Fig. R7e**).

Figure R7. Electrical current mapping measured by conductive atomic force microscopy of MAPbI₃ **a**, without, and with **b**, 5.0wt% of PC, **c**, 5.0wt% of PPC, **d**, 5.0wt% of PAA. **e**, current density vs. voltage (J-V) characteristics of perovskite solar cell without, and with 5.0wt% of PC, PPC, and PAA.

Unlike small molecular additives, polymers have higher molecular length and lower molecular chain mobility, which seems to be important factors to sustain suitable interactions with the perovskite precursors and mediate the perovskite crystallization. We found that PPC, with a strong electron donating nature, strongly interacted with the perovskite precursors to form a long-range macromolecular intermediate phase. The increased activation energy for crystallization enabled enlarged crystal grains to form and the low long chain mobility of the macromolecular intermediate phase allowed the polycrystals to be crosslinked with polymer-perovskite crystal composite bridges (**Fig. 3**). For the reason above, PPC and PAA had obviously different influences on the crystal growth and electrical properties of the perovskite films. Therefore, we believe that our inter-grain crosslinking method which involves a macromolecular polymer-perovskite composite did not lead to a severe electrical degradation of the perovskite, even when incorporated with high concentrations of the insulating additive

molecules. Simultaneously, these perovskite films possessed a combination of advantageous optoelectronic/ electrical properties and material/ device stability.

Revised parts in the manuscript)

Page 12 in the manuscript,

“The ability to form an appropriate interaction between the polymer and the perovskite precursors crucially affects the perovskite crystal growth and the resulting electrical properties of the perovskite film. The Lewis basicity and molecular dipole of a polymer’s repeating functional group also affects the formation of the macromolecular intermediate phase with the high dipole moment perovskite precursor CH_3NH_3^+ ($\mu= 2.3\text{D}$). We compared the effects of different polymer functional groups on the perovskite crystal growth and subsequent electrical properties (See supplementary information, Fig. S9-14) and observed that the polymeric Lewis base increased the electrical conductivity of the perovskite while in contrast, the polymeric acid severely interrupted the spatial electrical conductance of the perovskite grains, which indeed resulted in a huge drop in the photovoltaic performance of the devices with a high dose of the polymeric acid molecules.”

Page 11-17 in the supplementary information,

“Three kinds of commercially available polymers, polyacrylic acid (PAA), poly(4-vinylpyridine) (PVP), and polyurethane (PU) were used to comparison (Fig. S9a). All the polymers have lone pairs of electrons on nitrogen or oxygen along the polymer backbone, but different basicity and molecular dipole moments. Pyridine ($\text{C}_5\text{H}_5\text{N}$) and urea ($\text{CH}_4\text{N}_2\text{O}$) are the Lewis base functional units of PVP and PU, respectively, but the dipole moment of urea (4.56

D) is higher than that of pyridine (2.2 D). Compared to the reference device ($\text{CH}_3\text{NH}_3\text{PbI}_3$ without polymeric additives) fabricated in the same batch, the addition of PAA and PVP into the $\text{CH}_3\text{NH}_3\text{PbI}_3$ perovskite degraded its photovoltaic performance, while a small amount of PU increased the photovoltaic performance of the perovskite solar cells. The addition of PU mainly improved the open circuit voltage and fill factor, increasing the power conversion efficiency (PCE) of the $\text{CH}_3\text{NH}_3\text{PbI}_3$ perovskite solar cells, similar to the effect of adding PPC.

Figure S9. Photovoltaic performance of solar cells with different polymers. a, Chemical structure of polyacrylic acid (PAA), polyvinylpyridine (PVP), and polyurethane (PU), b, Current density versus voltage ($J-V$) characteristics of the best $\text{CH}_3\text{NH}_3\text{PbI}_3$ solar cells with various polymers, and c, photovoltaic parameters of $\text{CH}_3\text{NH}_3\text{PbI}_3$ perovskite solar cells with different kinds of polymers. (short-circuit current density (J_{sc}), open-circuit voltage (V_{oc}), fill factor (FF) and power conversion efficiency (PCE))

Figure S10. Photovoltaic performance of solar cells with polyurethane. a, photovoltaic parameters of $\text{CH}_3\text{NH}_3\text{PbI}_3$ perovskite solar cells as a function of concentration of PU added (short-circuit current density (J_{SC}), open-circuit voltage (V_{OC}), fill factor (FF) and power conversion efficiency (PCE)), **b**, current density versus voltage (J-V) characteristics of the best $\text{CH}_3\text{NH}_3\text{PbI}_3$ solar cells with 0.1wt% and 0.3wt% PU.

Figure S11. Photovoltaic performance of solar cells according to molecular concentration. Photovoltaic parameters of $\text{CH}_3\text{NH}_3\text{PbI}_3$ perovskite solar cells with PC, PPC, and PAA according to their molecular concentration. (short-circuit current density (J_{sc}), open-circuit voltage (V_{oc}), fill factor (FF) and power conversion efficiency (PCE)).

The different effects of the small molecular Lewis base (i.e., propylene carbonate (PC)), polymeric Lewis base (i.e., PPC), and polymeric acid (i.e., polyacrylic acid (PAA)) on the electrical properties of the films and subsequent photovoltaic performances of the devices were investigated. We examined the photovoltaic performances of the perovskite solar cells according to the molecular concentrations ranging from 0.1wt% to 5.0wt%. We observed that the small molecule (PC) and polymeric Lewis base (PPC) mainly improved the open circuit voltages (V_{oc}) and fill factors (FF) of the devices, which showed no noticeable decrease even at high concentrations (Fig. S11). In contrast, PAA gradually decreased the grain size with increasing amounts of PAA added (Fig. S12). All the $\text{CH}_3\text{NH}_3\text{PbI}_3$ grains were finely cleaved

into nanometer-sized grains upon the addition of 5.0wt% of PAA (Fig. S12c), which could be attributed to the chemical interaction between PAA and the perovskite precursors/ or solvent in the solution.

Figure S12. Surface Morphology of perovskite films with PAA. Atomic force microscopy images of $\text{CH}_3\text{NH}_3\text{PbI}_3$ film **a**, without, **b**, with 0.3wt% of PAA, and **c**, with 5.0wt% of PAA.

Figure S13. Schematic diagrams of free energy vs. cluster radius.

Per classical theory for homogeneous nucleation, the nucleation rate is described by using a critical free energy (ΔG_c), which represents the free energy required for nuclei to be stable without being dissolved in the solution and is the sum of the surface and bulk free energy of the nuclei. This critical free energy is defined as the activation energy for nucleation and used

to describe the nucleation rate using an Arrhenius type equation [*Chem. Rev.* **114**, 7610 (2014), *Small* **7**, 2685 (2011)]:

$$\frac{dN}{dt} = A \exp\left(-\frac{\Delta G_c}{k_B T}\right) \quad (1)$$

where t is time, N is number of nuclei, A is pre-exponential factor, k_B is Boltzmann's constant, T is temperature. The crystal free energy (ΔG_c) can be written as a function of surface energy γ , molar volume v , supersaturation of solution S , which produce a following equation,

$$\frac{dN}{dt} = A \exp\left(-\frac{16\pi\gamma^3 v^2}{3k_B^3 T^3 (\ln S)^2}\right)$$

Because the repeating functional group of PAA is a carboxylic acid (C_2H_3COOH) ($pK_a = 4.25$), the PAA likely interacted with the Lewis base solvents of the precursor solution (i.e., DM SO, and DMF), instead of forming an adduct with the Lewis acidic precursors, which possibly increased the saturation level of the precursors in the solution, resulting in a decreased activation energy for crystallization (**Fig. S13**). As a result, the fast crystallization formed a large number of nuclei, and subsequently small crystal grains, which could severely interrupt the inter-grain electrical coupling due to the insulating nature of PAA residing in between the small perovskite polycrystals.

The reduction of crystal grain size and electrical insulation can also be considered in the aspect of inhibitor molecules by the following model [*Theor. Found. Chem. Eng.* **42**, 179 (2008)],

$$w(C) \sim \begin{cases} w(0) \exp(-K_I C), & C < C_C \\ 0, & C > C_C \end{cases} \quad K_I = \frac{4K f^{\frac{1}{2}} \gamma V_m}{kT \sigma \pi^{\frac{1}{2}}}$$

where w is crystallization rate [m/s], C is the inhibitor concentration [m^{-3}], C_C is the critical value of C , K is the adsorption constant [m], K_I is the coefficient of inhibition effectiveness [m^3], f is the area occupied by the adsorbed impurity particle [m^2], γ is the growth step energy [$J m^{-2}$], V_m is the unit volume [m^3] of the crystal, and σ is the relative solution supersaturation [dimensionless]. The equation quantifies the influence of inhibitors on the crystallization rate.

If the PAA does not have meaningful chemical interaction with the precursors, this additive can be regarded as an inhibitor for crystal growth. As C increases, the crystallization rate exponentially decreases, thereby resulting in small grain sizes of crystals with inhibitor molecules in between the crystals.

Figure S14. Electrical properties of perovskite films. Electrical current mapping measured by conductive atomic force microscopy of CH₃NH₃PbI₃ **a**, without, and with **b**, 5.0wt% of PC, **c**, 5.0wt% of PPC, **d**, 5.0wt% of PAA. **e**, current density vs. voltage (J-V) characteristics of perovskite solar cell without, and with 5.0wt% of PC, PPC, and PAA.

The perovskite films were deposited on an ITO/ SnO₂ electron transporting layer. To analyze their charge carrier conducting characteristics, a positive bias was applied using a Sb-doped Si tip. Both the small molecular Lewis base and polymeric Lewis base resulted in the overall electrical enhancement of the perovskite films compared to that of the bare CH₃NH₃PbI₃ film. Particularly, the electrical conductivity in the grain boundaries was seen to significantly increase even though the film had high concentrations of the insulating organic molecules

(5.0wt%). The conductivity enhancements could be attributed to an enhanced charge carrier mobility due to the higher crystallinity and an increased charge carrier concentration due to a reduction in both the defect trapping sites and non-radiative recombination in the perovskite grains. In contrast, the $\text{CH}_3\text{NH}_3\text{PbI}_3$ film with 5.0wt% PAA incorporated exhibited electrically decoupled tiny grains and inhomogeneity in between the perovskite grains (Fig. S14d). As a result, the existence of the insulating PAA in the perovskite film severely disrupted the electrical properties of the solar cells, degrading its J_{SC} and FF significantly (Fig. S14e).”

2. The performance (J_{sc} , V_{oc} , FF, PCE) of the as-prepared cells is compared in Fig. 5. No description of the reference sample in Fig.5d was given. At least I did not find it.

Answer: The averaged photovoltaic performances of the reference cells (solar cells with bare MAPbI_3) were J_{SC} : $21.79 \pm 0.39 \text{ mA/cm}^2$, V_{OC} : $1.084 \pm 0.019 \text{ V}$, FF: $0.719 \pm 0.022\%$, and PCE: $16.97 \pm 0.44\%$. As the reviewer suggested, we added the description of the photovoltaic performances of the as-prepared reference samples in the manuscript and these are also summarized in Table 2.

Revised parts in the manuscript)

Page 13 in the manuscript,

“The devices incorporated a planar heterojunction structure with an architecture of ITO/ SnO_2 /perovskite/spiro-MeOTAD/Ag. The averaged photovoltaic performances of reference cells (solar cells with bare $\text{CH}_3\text{NH}_3\text{PbI}_3$) were J_{SC} : $21.79 \pm 0.39 \text{ mA/cm}^2$, V_{OC} : $1.084 \pm 0.019 \text{ V}$, FF: $0.719 \pm 0.022\%$, and PCE: $16.97 \pm 0.44\%$ (Figure 5d and Table 2).”

Table 2. Photovoltaic parameters of solar cells of bare CH₃NH₃PbI₃ and CH₃NH₃PbI₃ with Lewis bases.

	J_{sc} (mA/cm²)	V_{oc} (V)	FF	PCE (%) (Best PCE)
CH₃NH₃PbI₃	21.79±0.39	1.084±0.019	0.719±0.022	16.97±0.44 (17.88)
w/ EC	22.05±0.28	1.115±0.019	0.759±0.013	18.65±0.51 (19.46)
w/ PC	22.38±0.12	1.118±0.011	0.763±0.014	19.09±0.44 (19.63)
w/ PPC	22.34±0.28	1.129±0.008	0.767±0.013	19.35±0.43 (20.06)

3. One of the major reasons for the degradation of these perovskite solar cells is the migration of the Iodine. This has been shown in several publications, for example Journal of Physical Chemistry Letters 7, 5168 (2016), which should be cited. How the polymer can prevent the Iodine migration? The authors seem to claim that the migration of ions is reversible, but I not sure. Can the authors clarify this aspect?

4. The PCE reported in Fig.5d seem to me quite high, or too optimistic. One typical problem in measuring the PCE of such solar cells is their long relaxation times leading to a hysteresis of the J-V characteristic. (One reason is the migration of Iodine, but also other polarization mechanisms.) Many groups overestimated the PCE because their measurement was done with too short time steps and thus was influenced by transient currents. See for example Solar Energy Materials and Solar Cells 159, 197 (2017) and Solar Energy 173, 976 (2018). Can the authors show in the J-V characteristics of Fig.5f

both scan directions of the voltage, forward and reverse? Is the hysteresis area reduced by using the polymer?

Answer for (3) and (4): We additionally cited the references [*J. of Phys. Chem. Lett.* 7, 5168 (2016), *Sol. Energy Mater Sol. Cells* 159, 197 (2017), and *Solar Energy* 173, 976 (2018)] suggested by the reviewer, because we also used Ag electrode that can form AgI compound due to ion migration from the perovskite layer, possibly resulting in J-V hysteresis and PCE degradation. We also added the comment about the possible compound formation and how this is related to the performance degradation. We appreciate the reviewer's helpful suggestion. Additionally, to compare the ion's electromigration characteristics and to investigate the operational degradation induced by ion migration, we have conducted extra experiments on the ion migration activation energy, current-voltage hysteresis, and operational degradation on the perovskites.

By measuring the thermal conductance of the perovskite film without/ with the polymeric Lewis base, we investigated the activation energy for ion migration in MAPbI₃. The Nernst-Einstein relation ($\sigma(T) = \frac{\sigma_0}{T} \exp\left(\frac{-E_a}{kT}\right)$, where k is the Boltzmann constant and σ_0 is a constant) was used to calculate the activation energy of the mobile ions, and the lateral conduction configuration was employed for the measurement (**Figure R8 inset**). Our macromolecular adduct approach, which resultantly modified the perovskite crystallization, effectively increased the ion migration activation energy from 0.40 eV (bare MAPbI₃) to 0.53 eV (MAPbI₃ with PPC) (**Fig. R8**), which can be attributed to the reduction of the perovskite's charged defects (e.g., positively charged anion vacancies) at the grain boundaries due to the passivation effect of the polymeric Lewis bases that remained in between the perovskite grains (**Fig. R9**).

Figure R8. Temperature-dependent conductivity of MAPbI₃ film **a**, without and **b**, with PPC (inset: the schematic illustration of the lateral conduction device configuration.)

Figure R9. a, Photoluminescence (PL) (Inset: normalized PL spectra) and **b**, time-resolved PL spectra of perovskite films without and with Lewis bases (Inset: PL lifetimes fitted from the time-resolved PL spectra).

According to a previous literature [*Energy Environ. Sci.* 10, 604 (2017)] cited in the manuscript, the operational behavior of devices as a function of time can be classified into two regimes: 1) initial reversible loss and 2) permanent degradation (Fig. R10a). They claimed that the reversible loss regime of perovskite solar cells is caused by the migration of ions and ionic defects because the activation energy and time scale for cation migration are much larger and longer (hours scale) than halide migration (minutes scale) (Fig. R10b-d).

[Redacted]

Figure R10. a, Maximum power point tracking for three perovskite solar cells (Device C was cyclically tracked 4 times for 5 hours and it was left in the dark at open circuit in between the consecutive measurements.), Schematics of the evolution of the ion distribution within the perovskite layer sandwiched between the electron and hole selective contacts under solar cell working conditions: **b**, initial conditions, **c**, non-stabilized conditions on the timescale of minutes and **d**, the stabilized condition on the timescale of hours [*Energy Environ. Sci.* 10, 604 (2017)].

We included a mention on the reversible loss regime as related to halide ion migration as “Considering that the initial decay is related to the reversible migration of charged defects, the observed reduced initial decay can be attributed to a decreased defect density as a result of the PPC-induced crystal modification.” As the reviewer pointed out, this sentence can mislead readers, so we revised it in our revised manuscript to avoid misunderstanding. We appreciate the reviewer’s helpful comment.

For the reason above, the hysteresis of the devices with PPC decreased compared to that of bare MAPbI₃ (**Fig. R11**). The hysteresis index was calculated for standard devices with/without PPC based on the following relation, $HI = \frac{PCE_{OC \rightarrow SC} - PCE_{SC \rightarrow OC}}{PCE_{OC \rightarrow SC} + PCE_{SC \rightarrow OC}}$. Our J-V measurements were recorded at 0.1 V/s (between 1.2 V and -0.1 V with 65 data points and 0.2 s of delay time per point). The device with PPC does not exhibit a significant difference according to the current-voltage scanning direction as compared to those with bare MAPbI₃. The J-V behavior under a forward bias (i.e., from short-circuit to open-circuit) did not change significantly (Forward: $V_{OC} = 1.146$ V, $J_{SC} = 21.33$ mA/cm², FF = 74.47%, and PCE = 18.21%, and Reverse: $V_{OC} = 1.149$ V, $J_{SC} = 21.53$ mA/cm², FF = 77.17%, and PCE = 19.09%), resulting in a low *HI* ~2.36%. In contrast, the reference device showed a larger discrepancy with different J-V scan directions (Forward: $V_{OC} = 1.097$ V, $J_{SC} = 21.53$ mA/cm², FF = 68.51%, and PCE = 16.18%, and Reverse: $V_{OC} = 1.109$ V, $J_{SC} = 21.48$ mA/cm², FF = 72.97%, and PCE = 17.38%), resulting in a higher *HI* ~3.58% than that with PPC.

Figure R11. Current density versus voltage (J-V) characteristics of the CH₃NH₃PbI₃ solar cells according to scanning direction **a**, without, and **b**, with PPC. (Reverse: 1.2 V to -0.1V, Forward: -0.1 V to 1.2 V).

The reduced charged defect density and increased activation energy for ion migration with PPC were also effective in decreasing the extent of initial decay during the cell operation. We fabricated three different kinds of perovskite solar cells, 1) MAPbI₃, 2) MA_{0.9}FA_{0.1}PbI₃, and 3) FA_{0.98}CS_{0.02}PbI₃ with 1.67 mol% of PEA₂PbI₄, with/ or without PPC. Steady-state power conversion efficiencies (PCEs) of encapsulated devices were periodically measured with aging under continuous illumination (0.9 sun, 90±5 mW/cm²) under open-circuit condition. PPC effectively elongated all the operational stability of the perovskite solar cells regardless of the ‘A’-site cation composition. The device with the FACS-based perovskite maintained ~80% of initial PCE after aging for 500 h, while MAPbI₃ degraded to ~20%, which can be attributed to the better material stability against light/ high temperature and the higher tolerance for ion migration relative to that of MAPbI₃. All the perovskite solar cells fabricated without PPC showed an exponential initial decay, followed by linear, stabilized decay. The devices with PPC substantially reduced the amount of this initial decrease (**Fig. R12**, and **Table R1**), which could be attributed to a reduced charged defect formation, effective defect passivation, and higher activation energy for their migration with PPC. We also extracted the T₈₀ (time with which the PCE decays to 80% of its initial PCE) from the post-burn-in regime (linear decay regime) for all the solar cells fabricated in this work and summarized the data in Table R1. The FACS-based solar cells with PPC decreased its initial decay to 1.8%, and after 500h illumination, maintained 95.0% of initial PCE, from which the expected T₈₀ is ~10800.1 h.

Figure R12. PCE evolution under light illumination (1.5 AM) as a function of time of encapsulated solar cells without and with polymeric Lewis base (PPC).

Table R1. Initial decay and calculated T_{80} of solar cells under light illumination.

	Initial decay [%]	T_{80} [h]
MA	44.7	169.5
MA w/ PPC	29.5	391.0
MAFA	36.0	288.0
MAFA w/ PPC	16.2	929.6
FACs	10.4	420.8
FACs w/ PPC	1.8	10800.1

Revised parts in the manuscript)

Page 14-15 in the manuscript,

“To demonstrate the universality of our cross-linking approach, we examined the operational behavior of the devices based on various kinds of ‘A’-site cations (MA, MA_{0.9}FA_{0.1}, and FA_{0.98}CS_{0.02}). To evaluate their operational stability, all the devices were encapsulated under a nitrogen atmosphere and exposed to continuous illumination (90 ±10 mW, without UV filter) under open-circuit condition (Supplementary Fig. 26, 27, and Supplementary Table 4). PPC effectively elongated the operational lifetime of all the perovskite solar cells regardless of their ‘A’-site cation composition. All the devices without and with PPC showed rapid initial decay followed by slower degradation with an almost linear profile, but the devices with PPC demonstrated a less severe initial decay. Given that the initial rapid decay can be related to the migration of ion and charged defects⁴⁸, the reduced initial decay regime can be attributed to a decreased charged defect density and an increased activation energy for ion migration as a result of the PPC-induced crystallization. Severe ion migration in perovskite solar cells can result in J-V hysteresis, and along with the possible compound formation with the electrode, could accelerate the degradation of a device during operation⁴⁹⁻⁵¹. Indeed, PPC effectively increased the activation energy for ion migration in the perovskite (Supplementary Fig. 28), resulting in reduced hysteresis than that of bare CH₃NH₃PbI₃ (Supplementary Fig. 29). Furthermore, the addition of PPC slowed down the subsequent linear decay regime as well. From this, we extracted the T₈₀ (time with which the PCE decays to 80% of its initial value) from the post-burn-in regime (linear decay regime) for all the solar cells fabricated in this work and summarized this in Supplementary Table S4. All the calculated T₈₀ lifetimes of the devices with PPC were significantly elongated. Especially, the FACs-based solar cells with PPC decreased its initial decay to 1.8%, and after 500h illumination, maintained 95.0% of initial

PCE, from which the expected T_{80} is ~ 10800.1 h. The slower decay has been related to a permanent degradation of the perovskite layer accompanied by a chemical reaction and morphological change⁵². Therefore, the inter-grain cross-linking of the perovskite, induced by the macromolecular intermediate phase, could very well have retarded the irreversible degradation of the perovskite as well.”

Page 21 in the manuscript,

49. Nemnes, G. A. *et al.* Dynamic electrical behavior of halide perovskite based solar cells. *Sol. Energy Mater Sol. Cells* **159**, 197 (2017).
50. Nemnes, G. A. *et al.* How measurement protocols influence the dynamic J-V characteristics of perovskite solar cells: Theory and experiment. *Solar Energy* **173**, 976 (2018).
51. Besleaga, C. *et al.* Iodine migration and degradation of perovskite solar cells enhanced by metallic electrodes. *J. of Phys. Chem. Lett.* **7**, 5168 (2016).

Page 29-30 in the supplementary information,

The Nernst-Einstein relation ($\sigma(T) = \frac{\sigma_0}{T} \exp(\frac{-E_a}{kT})$, where k is the Boltzmann constant and σ_0 is a constant) was used to calculate the activation energy for ion migration, and the lateral conduction configuration was employed for the measurement (Fig. S28 inset). Our macromolecular adduct approach, which resultantly modified the perovskite crystallization, effectively increased the ion migration activation energy from 0.40 eV (bare $\text{CH}_3\text{NH}_3\text{PbI}_3$) to 0.53 eV ($\text{CH}_3\text{NH}_3\text{PbI}_3$ with PPC) (Fig S28), which can be attributed to the reduction of the

perovskite's charged defects (e.g., positively charged anion vacancies) at the grain boundaries due to the passivation effect of the polymeric Lewis bases that remained in between the perovskite grains, and this translated to a reduced J-V hysteresis of the solar cells with PPC.

Figure S28. Activation energy for ion migration. Temperature-dependent conductivity of CH₃NH₃PbI₃ film **a**, without and **b**, with PPC (inset: schematic illustration of the lateral conduction device configuration.)

Figure S29. J-V hysteresis of solar cells. Current density versus voltage (J-V) characteristics of the CH₃NH₃PbI₃ solar cells according to scanning direction **a**, without, and **b**, with PPC. (Reverse: 1.2 V to -0.1V, Forward: -0.1 V to 1.2 V).

5. Environmental stress test was executed by placing the sample in 70% humidity for 150 h and at elevated temperature (100 °C) in inert ambient (N₂) for 66 h and exposure to light (1.5 MO) for 2 h. The performance of the cells after the environmental test was not given. It was only shown that the PCC suffered from less degradation than EC and PC (suppl. Fig. S17). Even though that does indicate that PCC will work better than EC and PC, it says little about the absolute performance of the cell. While these tests may be suitable to demonstrate a difference in using the different additives, a better comparison to a reference cell is needed.

Answer) We appreciate the reviewer's constructive comments. The ultraviolet-visible absorption (UV-vis-Abs) and X-ray diffraction (XRD) results in the manuscript were tested on perovskite films to investigate the influences of external degradation factors on the perovskite film with or without the effect of the polymers and their crosslinking (**Fig. R13**). From the results above, we confirmed that the PPC-mediated defect passivation and the enlarged/ cross-linked crystals are much more resistant to extrinsic degradation factors, namely moisture, heat, and light. To prove the better performance of the devices with PPC against these extrinsic factors, we have additionally conducted environmental stability tests on solar cell devices. [ITO/ SnO₂/ Perovskite film] was exposed to moisture and high temperature to distinguish the degradation effects between the perovskite and the hole transporting layer (HTL) (p-doped spiro-MeOTAD) because of the well-known hygroscopic nature of the Li-salt in the HTL, and the low morphological stability under high temperatures significantly degrade the solar cell's performance [*J. Mater. Chem. A*, **6**, 2219 (2016), *Nat. Commun.* **7**, 11105 (2016)]. After aging of the [ITO/ SnO₂/ Perovskite film] under humid/ high temperature environments, spiro-MeOTAD and Ag were deposited on it as a HTL and electrode, respectively (**Fig. R14**).

Figure R13. X-ray diffraction spectra of bare $\text{CH}_3\text{NH}_3\text{PbI}_3$ and $\text{CH}_3\text{NH}_3\text{PbI}_3\cdot\text{PPC}$ (0.3 and 5.0 wt%) films against **a**, moisture (relative humidity: $70\pm 5\%$), **b**, light (1.5 AM) and **c**, heat (100°C).

Figure R14. Schematic illustration of moisture and thermal environmental stability test of perovskite solar cells.

The photovoltaic performances of bare MAPbI_3 exposed to a humid environment ($\text{RH } 70\pm 5\%$) significantly dropped. A 50 h exposure almost fully converted the MAPbI_3 to yellow colored PbI_2 (**Fig. R15 inset**), which can be attributed to a moisture-induced decomposition of MAPbI_3

according to the following reaction $\text{CH}_3\text{NH}_3\text{PbI}_3 \rightarrow \text{CH}_3\text{NH}_2 + \text{HI} + \text{PbI}_2$ [*J. Mater. Chem. A* **3**, 8970 (2015)]. Most of the bare MAPbI₃ devices exposed to moisture for 50 h did not work as normal photovoltaics, and the devices that worked exhibited very poor performances, with huge drops in J_{SC} and FF (50 h exposed bare MAPbI₃ device: V_{OC}: 0.949±0.06 V, J_{SC}: 9.85±1.34 mA/cm², FF: 32.5±2.4%, and PCE: 3.08±%0.81) (**Fig. R15**). In contrast, the cross-linked MAPbI₃ with PPC showed a much higher moisture resistance and still maintained relatively high photovoltaic performances even after 50 h exposure in a high moisture environment (50 h exposed PPC-MAPbI₃ device: V_{OC}= 1.088±0.02 V, J_{SC}: 20.43±0.28 mA/cm², FF: 72.3±1.8%, and PCE: 16.07±%0.67).

Figure R15. a, Photovoltaic parameters of MAPbI₃ perovskite solar cells according to exposure time against moisture (RH 70±5%) without/ with PPC (short-circuit current density (J_{SC}), open-circuit voltage (V_{OC}), fill factor (FF) and power conversion efficiency (PCE)), Current density versus voltage (J-V) characteristics of the best CH₃NH₃PbI₃ solar cells as a function of exposure time **b**, without, and **c**, with PPC.

We also proved the enhanced thermal stability of the PPC-MAPbI₃ devices under high temperature (150 °C) heating of the [ITO/ SnO₂/ Perovskite film] for 1h. The device corresponding to the MAPbI₃ film heated at this temperature showed a significant drop in its photovoltaic performances (V_{OC} : 0.980 ± 0.04 V, J_{SC} : 16.89 ± 2.37 mA/cm², FF: $54.7 \pm 4.0\%$, PCE: $9.31 \pm 2.02\%$) (**Fig. R16**). Compared to bare MAPbI₃, the PPC-MAPbI₃ cross-linked devices did not show a significant drop even at 150 °C heating (V_{OC} : 1.064 ± 0.07 V, J_{SC} : 21.34 ± 0.28 mA/cm², FF: $74.6 \pm 2.3\%$, PCE: $16.93 \pm 0.77\%$), and proved their better thermal stress tolerance as seen from the XRD and UV-vis-Abs results.

Figure R16. a, Photovoltaic parameters of MAPbI₃ perovskite solar cells according to exposure against high temperature (150 °C) without/ with PPC (short-circuit current density (J_{SC}), open-circuit voltage (V_{OC}), fill factor (FF) and power conversion efficiency (PCE)), Current density versus voltage (J-V) characteristics of the best CH₃NH₃PbI₃ solar cells **b**, without, and **c**, with PPC.

Revised parts in the manuscript)

Page 17 in the manuscript,

“With increasing amount of PPC added, the degradation of the $\text{CH}_3\text{NH}_3\text{PbI}_3$ films was further retarded, supporting the conclusion that the improved environmental stability originated from the addition of the PPC and their inter-grain cross-linking effect. **The photovoltaic performances of the cross-linked PPC- $\text{CH}_3\text{NH}_3\text{PbI}_3$ solar cells also exhibited a much superior resistance against the harsh environmental conditions (Supplementary Fig. 20-22), and thus demonstrates excellent practical environmental stability.”**

Page 22-24 in the supplementary information,

Figure S20. Schematic illustration of the environmental stability test on the perovskite solar cells

After the perovskite layers were exposed to a humid environment ($\text{RH } 70 \pm 5\%$) (Fig. S20), the photovoltaic performances of the bare $\text{CH}_3\text{NH}_3\text{PbI}_3$ solar cells dropped significantly, and the working devices exhibited very poor performances, mainly having huge drops in J_{SC} and FF (50 h

exposed bare MAPbI₃ device: V_{oc} : 0.949 ± 0.06 V, J_{sc} : 9.85 ± 1.34 mA/cm², FF: $32.5 \pm 2.4\%$, and PCE: $3.08 \pm 0.81\%$ (**Fig. S21**). In contrast, the cross-linked CH₃NH₃PbI₃ with PPC showed much higher moisture resistance and still maintained relatively high photovoltaic performances even after 50 h exposure in the high moisture environment (50 h exposed PPC- CH₃NH₃PbI₃ device: V_{oc} = 1.088 ± 0.02 V, J_{sc} : 20.43 ± 0.28 mA/cm², FF: $72.3 \pm 1.8\%$, and PCE: $16.07 \pm 0.67\%$).

Figure S21. Moisture stability of solar cells. **a**, Photovoltaic parameters of CH₃NH₃PbI₃ perovskite solar cells according to exposure time against moisture (RH 70±5%) without/ with PPC (short-circuit current density (J_{sc}), open-circuit voltage (V_{oc}), fill factor (FF) and power conversion efficiency (PCE)), Current density versus voltage (J-V) characteristics of the best CH₃NH₃PbI₃ solar cells as a function of exposure time **b**, without, and **c**, with PPC.

The high temperature heating of the bare CH₃NH₃PbI₃ also caused the photovoltaic performances of the devices to drop significantly (V_{oc} : 0.980 ± 0.04 V, J_{sc} : 16.89 ± 2.37 mA/cm², FF: $54.7 \pm 4.0\%$,

PCE: $9.31 \pm 2.02\%$) (Fig. S22). Compared to bare $\text{CH}_3\text{NH}_3\text{PbI}_3$, the PPC- $\text{CH}_3\text{NH}_3\text{PbI}_3$ cross-linked devices did not show any significant drop even after heated at $150\text{ }^\circ\text{C}$ heating (V_{OC} : $1.064 \pm 0.07\text{ V}$, J_{SC} : $21.34 \pm 0.28\text{ mA/cm}^2$, FF: $74.6 \pm 2.3\%$, PCE: $16.93 \pm 0.77\%$), and proved their better thermal stress tolerance as seen in the XRD and UV-vis-Abs results.

Figure S22. Thermal stability of solar cells. a, Photovoltaic parameters of $\text{CH}_3\text{NH}_3\text{PbI}_3$ perovskite solar cells according to exposure against high temperature ($150\text{ }^\circ\text{C}$) without/ with PPC (short-circuit current density (J_{SC}), open-circuit voltage (V_{OC}), fill factor (FF) and power conversion efficiency (PCE)), Current density versus voltage (J-V) characteristics of the best $\text{CH}_3\text{NH}_3\text{PbI}_3$ solar cells b, without, and c, with PPC.

Reviewer #3

Comments:

This work describes polymer-perovskite composite film for perovskite solar cell. The methodology is basically based on Lewis acid-base adduct approach. Poly(propylene carbonate) (PPC) was used for this study. DMSO has been usually used to form adduct but DMSO is finally removed to form perovskite. However, in this work PPC as a Lewis base was not be removed from perovskite but included in perovskite film. By doing so perovskite grain size was increased and stability was also significantly improved. Polymer-perovskite compisite approach is expected to be platform technology in perovskite solar cell research. Thus this work should be published without delay but after minor revision.

1. The title is not attractive. I suggest the following title. "Perovskite-polymer composite approach for long-term stable and high efficiency perovskite solar cell"

Answer) We appreciate the reviewer's constructive comments. By accepting the reviewer's suggestion, we modified the tile of our manuscript to "*Perovskite-Polymer Composite Cross-linker Approach for Highly-stable and Efficient Perovskite Solar Cells*".

2. It is required to describe J-V hysteresis for the perovskite-polymer composite based solar cell.

Answer) The hysteresis of the devices with PPC decreased compared to that of bare MAPbI₃ (**Fig. R1**). The hysteresis index was calculated for standard devices with/ without PPC based on the

following relation, $HI = \frac{PCE_{OC \rightarrow SC} - PCE_{SC \rightarrow OC}}{PCE_{OC \rightarrow SC} + PCE_{SC \rightarrow OC}}$. Our J-V measurements were recorded at 0.1 V/s (between 1.2 V and -0.1 V with 65 data points and 0.2 s of delay time per point). The device with PPC does not exhibit a significant difference according to the current-voltage scanning direction as compared to those with bare MAPbI₃. The J-V behavior under a forward bias (i.e., from short-circuit to open-circuit) did not change significantly (Forward: V_{OC} = 1.146 V, J_{SC} = 21.33 mA/cm², FF = 74.47%, and PCE = 18.21%, and Reverse: V_{OC} = 1.149 V, J_{SC} = 21.53 mA/cm², FF = 77.17%, and PCE = 19.09%), resulting in a low HI ~2.36%. In contrast, the reference device showed a larger discrepancy with different J-V scan directions (Forward: V_{OC} = 1.097 V, J_{SC} = 21.53 mA/cm², FF = 68.51%, and PCE = 16.18%, and Reverse: V_{OC} = 1.109 V, J_{SC} = 21.48 mA/cm², FF = 72.97%, and PCE = 17.38%), resulting in a higher HI ~3.58% than that with PPC.

Figure R1. Current density versus voltage (J-V) characteristics of the CH₃NH₃PbI₃ solar cells according to scanning direction **a**, without, and **b**, with PPC. (Reverse: 1.2 V to -0.1 V, Forward: -0.1 V to 1.2 V).

By measuring the thermal conductance of the perovskite film without/ with the polymeric Lewis base, we investigated the activation energy for ion migration in MAPbI₃. The Nernst-Einstein

relation ($\sigma(T) = \frac{\sigma_0}{T} \exp(\frac{-E_a}{kT})$, where k is the Boltzmann constant and σ_0 is a constant) was used to calculate the activation energy of the mobile ions, and the lateral conduction configuration was employed for the measurement (**Figure R2 inset**). Our macromolecular adduct approach, which resultantly modified the perovskite crystallization, effectively increased the ion migration activation energy from 0.40 eV (bare MAPbI₃) to 0.53 eV (MAPbI₃ with PPC) (Fig. R2), which can be attributed to the reduction of the perovskite's charged defects (e.g., positively charged anion vacancies) at the grain boundaries due to the passivation effect of the polymeric Lewis bases that remained in between the perovskite grains (**Fig. R3**).

Figure R2. Temperature-dependent conductivity of MAPbI₃ film **a**, without and **b**, with PPC (inset: the schematic illustration of the lateral conduction device configuration.)

Figure R3. a, Photoluminescence (PL) (Inset: normalized PL spectra) and **b**, time-resolved PL spectra of perovskite films without and with Lewis bases (Inset: PL lifetimes fitted from the time-resolved PL spectra).

Revised parts in the manuscript)

Page 30-31 in the supplementary information,

The Nernst-Einstein relation ($\sigma(T) = \frac{\sigma_0}{T} \exp(\frac{-E_a}{kT})$, where k is the Boltzmann constant and σ_0 is a constant) was used to calculate the activation energy for ion migration, and the lateral conduction configuration was employed for the measurement (Fig. S28 inset). Our macromolecular adduct approach, which resultantly modified the perovskite crystallization, effectively increased the ion migration activation energy from 0.40 eV (bare $\text{CH}_3\text{NH}_3\text{PbI}_3$) to 0.53 eV ($\text{CH}_3\text{NH}_3\text{PbI}_3$ with PPC) (Fig S28), which can be attributed to the reduction of the perovskite's charged defects (e.g., positively charged anion vacancies) at the grain boundaries due to the passivation effect of the polymeric Lewis bases that remained in between the perovskite grains, and this translated to a reduced J-V hysteresis of the solar cells with PPC.

Figure S28. Activation energy for ion migration. Temperature-dependent conductivity of $\text{CH}_3\text{NH}_3\text{PbI}_3$ film **a**, without and **b**, with PPC (inset: schematic illustration of the lateral conduction device configuration.)

Figure S29. J-V hysteresis of solar cells. Current density versus voltage (J-V) characteristics of the $\text{CH}_3\text{NH}_3\text{PbI}_3$ solar cells according to scanning direction **a**, without, and **b**, with PPC. (Reverse: 1.2 V to -0.1 V, Forward: -0.1 V to 1.2 V).

3. Except for PPC, it is necessary to suggest other candidates (hydrophobic polymers) for perovskite-polymer composite approach.

Answer) As the reviewer suggested, we additionally compared the performances of perovskite solar cells incorporated with additives with different functional repeating groups, which can suggest new guidelines related to the functional group requirements of the polymers. We used three additional kinds of commercially available polymers, polyacrylic acid (PAA), poly(4-vinylpyridine) (PVP), and polyurethane (PU) (**Fig. R4a**). All the polymers have lone pairs of electrons on nitrogen or oxygen along the polymer backbone, but different basicity and molecular dipole moments. Acrylic acid, the repeating unit of PAA, is the simplest unsaturated carboxylic

acid (C_2H_3COOH) ($pK_a= 4.25$) with a molecular dipole moment of 1.46 D. Pyridine (C_5H_5N) and urea (CH_4N_2O) are the Lewis base functional units of PVP and PU, respectively, but the dipole moment of urea (4.56 D) is higher than that of pyridine (2.2 D). In other words, we chose three different polymers representative of a polymeric acid (PAA), a Lewis base with high (PU) and low (PVP) molecular dipole moment functional groups. As shown in Figure R1b and c, compared to the reference device (MAPbI₃ without polymeric additives) fabricated in the same batch, the addition of PAA and PVP into the MAPbI₃ perovskite degraded its photovoltaic performance, while a small amount of PU increased the photovoltaic performance of the perovskite solar cells. The addition of PU mainly improved the open circuit voltage (V_{OC}) to 1.155 V and fill factor (FF) to 71.3%, increasing the power conversion efficiency (PCE) of the MAPbI₃ perovskite solar cells (from 16.5% to 17.7% PCE), similar to the effect of adding PPC.

Figure R4. a, Chemical structure of polyacrylic acid (PAA), polyvinylpyridine (PVP), and polyurethane (PU), **b**, Current density versus voltage (J-V) characteristics of the best CH₃NH₃PbI₃ solar cells with various polymers, and **c**, photovoltaic parameters of CH₃NH₃PbI₃ perovskite solar cells with different kinds of polymers. (short-circuit current density (J_{SC}), open-circuit voltage (V_{OC}), fill factor (FF) and power conversion efficiency (PCE))

As the PU concentration increased (0.1wt% and 0.3wt%), the V_{OC} and FF increased, while the J_{SC} showed no noticeable decrease, resulting in an improved PCE (from 16.5% to 17.7%) (**Fig. R5**). In contrast, the J_{SC} and FF severely decreased (J_{SC}: 21.3 mA/cm² to 17.2 mA/cm², FF: 69.6% to 58.7%) as the PAA additive concentration in the perovskite precursor solution was increased (**Fig. R6**), which means that PAA significantly degraded the electrical properties of MAPbI₃. The ability to form an appropriate interaction between the polymer and the perovskite precursors crucially affects the perovskite crystal growth and the resulting electrical properties of the perovskite film. The Lewis basicity and molecular dipole of a polymer's repeating functional group also affects the formation of the macromolecular intermediate phase with the high dipole moment perovskite precursor CH₃NH₃⁺ ($\mu = 2.3\text{D}$). Also, as the reviewer suggested, the hydrophobicity of the polymer chain could be one additional requirement to enhance the environmental/ operational stability of the solar cells (Surface energy of PU is ~ 32 dynes/cm, which is relatively hydrophobic nature).

Figure R5. a, photovoltaic parameters of $\text{CH}_3\text{NH}_3\text{PbI}_3$ perovskite solar cells as a function of concentration of PAA added (short-circuit current density (J_{SC}), open-circuit voltage (V_{OC}), fill factor (FF) and power conversion efficiency (PCE)), **b**, current density versus voltage (J-V) characteristics of the best $\text{CH}_3\text{NH}_3\text{PbI}_3$ solar cells with 0.1wt% and 0.3wt% PAA.

Figure R6. a, photovoltaic parameters of $\text{CH}_3\text{NH}_3\text{PbI}_3$ perovskite solar cells as a function of concentration of PU added (short-circuit current density (J_{sc}), open-circuit voltage (V_{oc}), fill factor (FF) and power conversion efficiency (PCE)), **b**, current density versus voltage (J-V) characteristics of the best $\text{CH}_3\text{NH}_3\text{PbI}_3$ solar cells with 0.1wt% and 0.3wt% PU.

Revised parts in the manuscript)

Page 10-11 in the manuscript,

“The ability to form an appropriate interaction between the polymer and the perovskite precursors crucially affects the perovskite crystal growth and the resulting electrical properties of the perovskite film. The Lewis basicity and molecular dipole of a polymer’s repeating functional group also affects the formation of the macromolecular intermediate phase with the high dipole moment perovskite precursor CH_3NH_3^+ ($\mu= 2.3\text{D}$). We compared the effects of different polymer functional groups on the perovskite crystal growth and subsequent electrical properties (See supplementary information, Fig. S9-14) and observed that the polymeric Lewis base increased the electrical conductivity of the perovskite while in contrast, the polymeric acid severely interrupted the spatial electrical conductance of the perovskite grains, which indeed resulted in a huge drop in the photovoltaic performance of the devices with a high dose of the polymeric acid molecules.”

Page 11-13 in the supplementary information,

“Three kinds of commercially available polymers, polyacrylic acid (PAA), poly(4-vinylpyridine) (PVP), and polyurethane (PU) were used to comparison (Fig. S9a). All the polymers have lone pairs of electrons on nitrogen or oxygen along the polymer backbone, but different basicity and molecular dipole moments. Pyridine ($\text{C}_5\text{H}_5\text{N}$) and urea ($\text{CH}_4\text{N}_2\text{O}$) are the Lewis base functional

units of PVP and PU, respectively, but the dipole moment of urea (4.56 D) is higher than that of pyridine (2.2 D). Compared to the reference device ($\text{CH}_3\text{NH}_3\text{PbI}_3$ without polymeric additives) fabricated in the same batch, the addition of PAA and PVP into the $\text{CH}_3\text{NH}_3\text{PbI}_3$ perovskite degraded its photovoltaic performance, while a small amount of PU increased the photovoltaic performance of the perovskite solar cells. The addition of PU mainly improved the open circuit voltage and fill factor, increasing the power conversion efficiency (PCE) of the $\text{CH}_3\text{NH}_3\text{PbI}_3$ perovskite solar cells, similar to the effect of adding PPC.

Figure S9. Photovoltaic performance of solar cells with different polymers. **a**, Chemical structure of polyacrylic acid (PAA), polyvinylpyridine (PVP), and polyurethane (PU), **b**, Current density versus voltage (J - V) characteristics of the best $\text{CH}_3\text{NH}_3\text{PbI}_3$ solar cells with various polymers, and **c**, photovoltaic parameters of $\text{CH}_3\text{NH}_3\text{PbI}_3$ perovskite solar cells with different kinds of polymers. (short-circuit current density (J_{sc}), open-circuit voltage (V_{oc}), fill factor (FF) and power conversion efficiency (PCE))

Figure S10. Photovoltaic performance of solar cells with polyurethane. a, photovoltaic parameters of CH₃NH₃PbI₃ perovskite solar cells as a function of concentration of PU added (short-circuit current density (J_{sc}), open-circuit voltage (V_{oc}), fill factor (FF) and power conversion efficiency (PCE)), **b,** current density versus voltage (J-V) characteristics of the best CH₃NH₃PbI₃ solar cells with 0.1wt% and 0.3wt% PU.”

4. Since spiro-MeOTAD was used, thermal stability may have problem?

Answer: As the reviewer commented, because of the well-known hygroscopic nature of the Li-salt in the HTL, and the low morphological stability under high temperatures significantly degrade the solar cell’s performance [[*J. Mater. Chem. A*, **6**, 2219 (2016), *Nat. Commun.* **7**, 11105 (2016)]. Therefore, we conducted additional experiments on the moisture and thermal stability of the perovskite solar cells. As seen in Figure 5, we confirmed that the PPC-mediated defect passivation and the enlarged/ cross-linked crystals are much more resistant to extrinsic degradation factors,

namely moisture, heat, and light. To prove the better performance of the devices with PPC against these extrinsic factors, we have additionally conducted environmental stability tests on solar cell devices. [ITO/ SnO₂/ Perovskite film] was exposed to moisture and high temperature to distinguish the degradation effects between the perovskite and the hole transporting layer (HTL) (p-doped spiro-MeOTAD) (**Fig. R7**).

Figure R7. Schematic illustration of moisture and thermal environmental stability test of perovskite solar cells.

The photovoltaic performances of bare MAPbI₃ exposed to a humid environment (RH 70±5%) significantly dropped. A 50 h exposure almost fully converted the MAPbI₃ to yellow colored PbI₂ (**Fig. R8 inset**), which can be attributed to a moisture-induced decomposition of MAPbI₃ according to the following reaction $\text{CH}_3\text{NH}_3\text{PbI}_3 \rightarrow \text{CH}_3\text{NH}_2 + \text{HI} + \text{PbI}_2$ [*J. Mater. Chem. A* **3**, 8970 (2015)]. Most of the bare MAPbI₃ devices exposed to moisture for 50 h did not work as normal photovoltaics, and the devices that worked exhibited very poor performances, with huge drops in J_{SC} and FF (50 h exposed bare MAPbI₃ device: V_{OC}: 0.949±0.06 V, J_{SC}: 9.85±1.34 mA/cm², FF: 32.5±2.4%, and PCE: 3.08±%0.81) (**Fig. R8**). In contrast, the cross-linked MAPbI₃

with PPC showed a much higher moisture resistance and still maintained relatively high photovoltaic performances even after 50 h exposure in a high moisture environment (50 h exposed PPC-MAPbI₃ device: $V_{OC}= 1.088\pm 0.02$ V, $J_{SC}: 20.43\pm 0.28$ mA/cm², FF: $72.3\pm 1.8\%$, and PCE: $16.07\pm 0.67\%$).

Figure R8. a, Photovoltaic parameters of MAPbI₃ perovskite solar cells according to exposure time against moisture (RH 70±5%) without/ with PPC (short-circuit current density (J_{SC}), open-circuit voltage (V_{OC}), fill factor (FF) and power conversion efficiency (PCE)), Current density versus voltage (J-V) characteristics of the best CH₃NH₃PbI₃ solar cells as a function of exposure time b, without, and c, with PPC.

We also proved the enhanced thermal stability of the PPC-MAPbI₃ devices under high temperature (150 °C) heating of the [ITO/ SnO₂/ Perovskite film] for 1h. The device corresponding to the MAPbI₃ film heated at this temperature showed a significant drop in its photovoltaic

performances (V_{oc} : 0.980 ± 0.04 V, J_{sc} : 16.89 ± 2.37 mA/cm², FF: $54.7 \pm 4.0\%$, PCE: $9.31 \pm 2.02\%$) (Fig. R9). Compared to bare MAPbI₃, the PPC-MAPbI₃ cross-linked devices did not show a significant drop even at 150 °C heating (V_{oc} : 1.064 ± 0.07 V, J_{sc} : 21.34 ± 0.28 mA/cm², FF: $74.6 \pm 2.3\%$, PCE: $16.93 \pm 0.77\%$), and proved their better thermal stress tolerance as seen from the XRD and UV-vis-Abs results.

Figure R9. a, Photovoltaic parameters of MAPbI₃ perovskite solar cells according to exposure against high temperature (150 °C) without/ with PPC (short-circuit current density (J_{sc}), open-circuit voltage (V_{oc}), fill factor (FF) and power conversion efficiency (PCE)), Current density versus voltage (J-V) characteristics of the best CH₃NH₃PbI₃ solar cells b, without, and c, with PPC.

Revised parts in the manuscript)

Page 13 in the manuscript,

“With increasing amount of PPC added, the degradation of the $\text{CH}_3\text{NH}_3\text{PbI}_3$ films was further retarded, supporting the conclusion that the improved environmental stability originated from the addition of the PPC and their inter-grain cross-linking effect. **The photovoltaic performances of the cross-linked PPC- $\text{CH}_3\text{NH}_3\text{PbI}_3$ solar cells also exhibited a much superior resistance against the harsh environmental conditions (Supplementary Fig. 20-22), and thus demonstrates excellent practical environmental stability.”**

Page 22-24 in the supplementary information,

Figure S20. Schematic illustration of the environmental stability test on the perovskite solar cells

After the perovskite layers were exposed to a humid environment ($\text{RH } 70 \pm 5\%$) (**Fig. S20**), the photovoltaic performances of the bare $\text{CH}_3\text{NH}_3\text{PbI}_3$ solar cells dropped significantly, and the

working devices exhibited very poor performances, mainly having huge drops in J_{SC} and FF (50 h exposed bare MAPbI₃ device: V_{OC} : 0.949 ± 0.06 V, J_{SC} : 9.85 ± 1.34 mA/cm², FF: $32.5 \pm 2.4\%$, and PCE: $3.08 \pm 0.81\%$) (Fig. S21). In contrast, the cross-linked CH₃NH₃PbI₃ with PPC showed much higher moisture resistance and still maintained relatively high photovoltaic performances even after 50 h exposure in the high moisture environment (50 h exposed PPC- CH₃NH₃PbI₃ device: V_{OC} = 1.088 ± 0.02 V, J_{SC} : 20.43 ± 0.28 mA/cm², FF: $72.3 \pm 1.8\%$, and PCE: $16.07 \pm 0.67\%$).

Figure S21. Moisture stability of solar cells. **a**, Photovoltaic parameters of CH₃NH₃PbI₃ perovskite solar cells according to exposure time against moisture (RH 70±5%) without/ with PPC (short-circuit current density (J_{SC}), open-circuit voltage (V_{OC}), fill factor (FF) and power conversion efficiency (PCE)), Current density versus voltage (J-V) characteristics of the best CH₃NH₃PbI₃ solar cells as a function of exposure time **b**, without, and **c**, with PPC.

The high temperature heating of the bare $\text{CH}_3\text{NH}_3\text{PbI}_3$ also caused the photovoltaic performances of the devices to drop significantly (V_{OC} : 0.980 ± 0.04 V, J_{SC} : 16.89 ± 2.37 mA/cm², FF: $54.7 \pm 4.0\%$, PCE: $9.31 \pm 2.02\%$) (Fig. S22). Compared to bare $\text{CH}_3\text{NH}_3\text{PbI}_3$, the PPC- $\text{CH}_3\text{NH}_3\text{PbI}_3$ cross-linked devices did not show any significant drop even after heated at 150 °C heating (V_{OC} : 1.064 ± 0.07 V, J_{SC} : 21.34 ± 0.28 mA/cm², FF: $74.6 \pm 2.3\%$, PCE: $16.93 \pm 0.77\%$), and proved their better thermal stress tolerance as seen in the XRD and UV-vis-Abs results.

Figure S22. Thermal stability of solar cells. a, Photovoltaic parameters of $\text{CH}_3\text{NH}_3\text{PbI}_3$ perovskite solar cells according to exposure against high temperature (150 °C) without/ with PPC (short-circuit current density (J_{SC}), open-circuit voltage (V_{OC}), fill factor (FF) and power conversion efficiency (PCE)), Current density versus voltage (J-V) characteristics of the best $\text{CH}_3\text{NH}_3\text{PbI}_3$ solar cells b, without, and c, with PPC.

REVIEWERS' COMMENTS:

Reviewer #1 (Remarks to the Author):

The authors fully addressed my concerns and now the manuscript can be accepted.

Reviewer #2 (Remarks to the Author):

The authors made the changes recommended by the reviewer(s) in a satisfactory manner and the paper can be published.

Reviewer #3 (Remarks to the Author):

The manuscript was revised accordingly by answering all the questions raised by the reviewers. The corrections sound reasonable, thus this work is now suitable for publication without further revision.